# Physics-informed neural networks for solving nonlinear diffusivity and Biot's equations

**Teeratorn Kadeethum[1,2]\*, Thomas M. Jørgensen[1], Hamidreza M. Nick[2]**

**1** Department of Applied Mathematics and Computer Science, Technical University of Denmark, Lyngby, Denmark, **2** The Danish Hydrocarbon Research and Technology Centre, Technical University of Denmark, Lyngby, Denmark

\* teekad@dtu.dk

**Data Availability Statement:** The data used as train, validation, and test sets is provided in https://github.com/teeratornk/PINN_data.

**Funding:** The research leading to these results has received funding from the Danish Hydrocarbon

## Abstract

This paper presents the potential of applying physics-informed neural networks for solving nonlinear multiphysics problems, which are essential to many fields such as biomedical engineering, earthquake prediction, and underground energy harvesting. Specifically, we investigate how to extend the methodology of physics-informed neural networks to solve both the forward and inverse problems in relation to the nonlinear diffusivity and Biot's equations. We explore the accuracy of the physics-informed neural networks with different training example sizes and choices of hyperparameters. The impacts of the stochastic variations between various training realizations are also investigated. In the inverse case, we also study the effects of noisy measurements. Furthermore, we address the challenge of selecting the hyperparameters of the inverse model and illustrate how this challenge is linked to the hyperparameters selection performed for the forward one.

## Introduction

The volumetric displacement of a porous medium caused by the changes in fluid pressure inside the pore spaces is essential for many applications, including groundwater flow, underground heat mining, fossil fuel production, earthquake mechanics, and biomedical engineering [1–5]. Such volumetric deformation may impact the hydraulic storability and permeability of porous material, which influences the fluid flow behavior. This multiphysics problem, i.e., the coupling between fluid flow and solid deformation, can be captured through the application of Biot's equations of poroelasticity [6, 7]. The Biot's equations can be solved by analytical solutions for simple cases [8, 9]. In complex cases, the finite difference approximation [10, 11], finite volume discretization [12, 13], or more commonly finite element methods such as the mixed formulation or discontinuous/enriched Galerkin, [14–21], can be used. These numerical methods, however, require significant computational resources, and the accuracy of the solution depends heavily on the quality of the generated mesh. Hence, these methods may not be suitable to handle an inverse problem [22, 23] or a forward problem with complex geometries [24–26].

Recent proposals have speculated that neural-network-based approaches such as deep learning might be an appealing alternative in solving physical problems, which are governed

Research and Technology Centre under the Advanced Water Flooding program. The funders had no role in study design, data collection and analysis, decision to publish, or preparation of the manuscript.

**Competing interests:** The authors have declared that no competing interests exist.

by partial differential equations since they operate without a mesh and are scalable on multi-threaded systems [25–29]. Moreover, deep learning has been successfully applied to many applications [30–32] because of its capability to handle highly nonlinear problems [33]. As this technique, in general, requires a significantly large data set to reach a reasonable accuracy [34], its applicability to many scientific and industrial problems can be a challenge [35]. Use of a priori knowledge, however, can reduce the amount of examples needed. For instance, the idea of encoding physical information into the architectures and loss functions of the deep neural networks has been successfully applied to computational fluid dynamics problems [24, 26–29, 36]. The published results illustrate that by incorporating the physical information in the form of regularization terms of the loss functions, the neural networks can be trained to provide good accuracy with a reasonably sized data set.

Since the coupled fluid and solid mechanics process is highly nonlinear and generally involves complex geometries [19, 37, 38], it seems to fit well into the context of physics-informed neural networks (PINN). For this reason, we propose to apply PINN for solving the nonlinear diffusivity and Biot's equations, both concerning forward and inverse modeling. The rest of the paper is organized as follows. The governing equations of the coupled solid and fluid mechanics are presented in the methodology section. Subsequently, the PINN architecture and its loss functions are defined. We then present forward and inverse modeling results for both the nonlinear diffusivity equation as well as Biot's equations. We also study the impact of the stochastic variations of the training procedures on the accuracy of the predicted primary variables and estimated parameters. Finally, we conclude the findings and describe the possibilities that can enhance the capability of this model, which should be addressed in future works.

## Methodology

### Governing equations

We are interested in solving the nonlinear Biot's equations on the closed domain ($\Omega$) which amounts to a time-dependent multiphysics problem coupling solid deformation with fluid flow. Let $\Omega \subset \mathbb{R}^d$ be the domain of interest in $d$-dimensional space where $d = 1, 2,$ or $3$ and bounded by boundary, $\partial\Omega$. $\partial\Omega$ can be decomposed into displacement and traction boundaries, $\partial\Omega_u$ and $\partial\Omega_\sigma$, respectively, for the solid deformation problem. For the fluid flow problem, $\partial\Omega$ is decomposed into pressure and flux boundaries, $\partial\Omega_p$ and $\partial\Omega_q$, respectively. In short, $\partial\Omega_u$ and $\partial\Omega_p$ represent the first-kind boundary condition or Dirichlet boundary condition ($\partial\Omega_D$). The $\partial\Omega_\sigma$ and $\partial\Omega_q$, on the other hand, represent the second-kind boundary condition or Neumann boundary condition ($\partial\Omega_N$). The time domain is denoted by $\mathbb{T} = (0, \tau]$ with $\tau > 0$.

As just stated, the coupling between the fluid flow and solid deformation can be captured through the application of Biot's equations of poroelasticity, which is composed of linear momentum and mass balance equations [6]. The linear momentum balance equation can be written as follows:

$$\nabla \cdot \boldsymbol{\sigma}(\boldsymbol{u}, p) = \boldsymbol{f}, \tag{1}$$

where $\boldsymbol{u}$ is displacement, $p$ is fluid pressure, $\boldsymbol{f}$ is body force. The bold-face letters or symbols denote tensors and vectors, and the normal letters or symbols denote scalar values. Here, $\boldsymbol{\sigma}$ is the total stress, which is defined as:

$$\boldsymbol{\sigma} := \boldsymbol{\sigma}'(\boldsymbol{u}) - \alpha p \boldsymbol{I}, \tag{2}$$

where $I$ is the identity tensor and $\alpha$ is Biot's coefficient defined as [39]:

$$\alpha := 1 - \frac{K}{K_s},\tag{3}$$

with the bulk modulus of a rock matrix $K$ and the solid grains modulus $K_s$. In addition, $\boldsymbol{\sigma}'$ is an effective stress defined as:

$$\boldsymbol{\sigma}'(\boldsymbol{u}) := 2\mu_l\varepsilon(\boldsymbol{u}) + \lambda_l u\boldsymbol{I},\tag{4}$$

where $\lambda_l$ and $\mu_l$ are Lamé constants, $\varepsilon(\boldsymbol{u})$ is strain assuming infinitesimal displacements defined as:

$$\varepsilon(\boldsymbol{u}) := \frac{1}{2}(\nabla u + \nabla^T u).\tag{5}$$

We can write the linear momentum balance and its boundary conditions as:

$$\nabla \cdot \boldsymbol{\sigma}'(\boldsymbol{u}) - \alpha\nabla \cdot p\boldsymbol{I} = \boldsymbol{f} \text{ in } \Omega \times \mathbb{T},\tag{6}$$

$$\boldsymbol{u} = \boldsymbol{u}_D \text{ on } \partial\Omega_u \times \mathbb{T},\tag{7}$$

$$\boldsymbol{\sigma} \cdot \boldsymbol{n} = \boldsymbol{\sigma}_D \text{ on } \partial\Omega_\sigma \times \mathbb{T},\tag{8}$$

$$\boldsymbol{u} = \boldsymbol{u}_0 \text{ in } \Omega \text{ at } t = 0,\tag{9}$$

where $\boldsymbol{u}_D$ and $\boldsymbol{\sigma}_D$ are prescribed displacement and traction at boundaries, respectively, $\boldsymbol{n}$ is a normal unit vector, and $t$ is time.

The mass balance equation is written as [38, 40]:

$$\left(\phi c_f + \frac{\alpha - \phi}{K_s}\right)\frac{\partial p}{\partial t} + \alpha\frac{\partial\nabla \cdot u}{\partial t} - \nabla \cdot \mathcal{N}[\boldsymbol{\kappa}](\nabla p - \rho\mathbf{g}) = g \text{ in } \Omega \times \mathbb{T},\tag{10}$$

$$p = p_D \text{ on } \partial\Omega_p \times \mathbb{T},\tag{11}$$

$$-\mathcal{N}[\boldsymbol{\kappa}](\nabla p - \rho\mathbf{g}) \cdot \boldsymbol{n} = q_D \text{ on } \partial\Omega_q \times \mathbb{T},\tag{12}$$

$$p = p_0 \text{ in } \Omega \text{ at } t = 0,\tag{13}$$

where $\rho$ is fluid density, $\phi$ is initial porosity and remains constant throughout the simulation (the volumetric deformation is represented by $\partial\nabla \cdot \boldsymbol{u}/\partial t$), $c_f$ is fluid compressibility, $\mathbf{g}$ is a gravitational vector, $g$ is sink/source, $p_D$ and $q_D$ are specified pressure and flux, respectively, $\mathcal{N}[\cdot]$ represents a nonlinear operator, and $\boldsymbol{\kappa}$ is hydraulic conductivity defined as:

$$\boldsymbol{\kappa} := \begin{bmatrix} \kappa^{xx} & \kappa^{xy} & \kappa^{xz} \\ \kappa^{yx} & \kappa^{yy} & \kappa^{yz} \\ \kappa^{zx} & \kappa^{zy} & \kappa^{zz} \end{bmatrix},\tag{14}$$

where the tensor components characterize the transformation of the components of the gradient of fluid pressure into the components of the velocity vector. The $\kappa^{xx}$, $\kappa^{yy}$, and $\kappa^{zz}$ represent the matrix permeability in x-, y-, and z-direction, respectively. In this study, all off-diagonal terms are zero because we assume that a porous media is isotropic and homogeneous [41, 42].

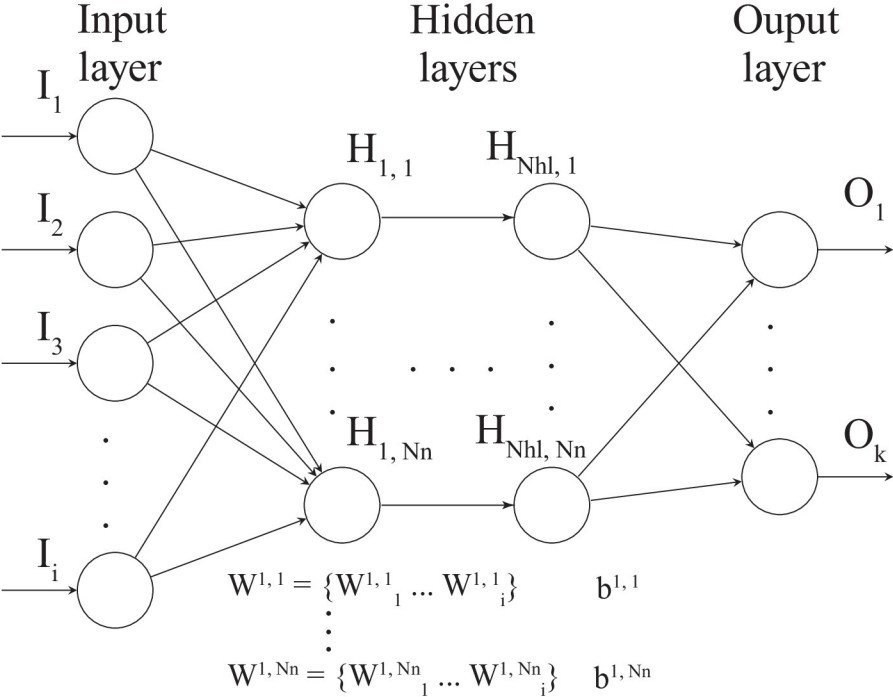

**Fig 1. General neural network architecture used in this study [33, 43, 44].** The input layer contains up to $i$ input nodes, and the output layer is composed of $1, \ldots, k$ output nodes. $N_{hl}$ refes to the number of hidden layers, and each hidden layer is composed of $N_n$ neurons. Each neuron (e.g., $H_{1,1} \ldots H_{1,N_n}$) is connected to the nodes of the previous layer with adjustable weights and also has an adjustable bias.

Note that the diffusivity equation is a specific case of the Biot's equations since Eqs (6) and (10) decouple when $\alpha = 0$. The details of this equation are presented in the results and discussion section.

## Physics-informed neural networks model

**Neural network architecture.** The neural network architecture used in this study is presented in Fig 1 [33, 43, 44]. The number of input and output nodes in the neural networks are determined from the problem formulation; for example, if the problem is time-dependent poroelasticity (as discussed in the previous section) and bounded by $\Omega = [0, 1]^1$, we have two input nodes ($x$ and $t$) and two output nodes ($u$ and $p$) where $x$ is coordinate in x-direction, $t$ is time, $u$ is the displacement in x-direction, and $p$ is fluid pressure. The number of hidden layers ($N_{hl}$) and the number of neurons ($N_n$) act as so-called hyperparameters [45]. Each neuron (e.g., $H_{1,1} \ldots H_{1,N_n}$) is connected to the nodes of the previous layer with adjustable weights and also has an adjustable bias. We denote the set of weights and biases as (W) and (b), respectively. These variables are learned during a training phase [44, 45]. In this paper, we define all hidden layers to have the same number of neurons.

Physics-informed neural networks encode the information given by the differential operators as specific regularizing terms of the loss functions used when training the networks (see the section below). Because the training examples that are used to evaluate these extra regularizing terms, in general, are different from those used to train the network shown in Fig 1, one conceptually introduces an additional neural network—denoted the physical informed neural network [26]. This additional neural network is dependent on all the W and b of the first

neural network, but it introduces some extra variables to be learned for the inverse modeling case. This is elaborated in more detail in the below section on training the PINN. Note that while the neural network shown in Fig 1 can be seen as point-to-point learning, the physics-informed regularization indirectly contributes to the local interactions (the stencil). The neural networks are built on the Tensorflow platform [46]. The results produced using either the rectified linear unit (ReLU) or the hyperbolic tangent (tanh) were comparable. Hence, we only present the results using the tanh activation function in this paper.

**Physics-informed function.**   We encode the underlying physical information to the neural networks through the so-called physics-informed function ($\Pi$), acting as additional regularizing terms in the loss function defined below.

For the linear momentum balance equation Eq (6), we define $\Pi_u$ as follows:

$$\Pi_u := \nabla \cdot \boldsymbol{\sigma}'(\boldsymbol{u}) - \alpha \nabla \cdot p\boldsymbol{I} - \boldsymbol{f} \ \text{ in } \ \Omega \times \mathbb{T}, \tag{15}$$

and with reference to Eq (8) for its $\partial\Omega_\sigma$:

$$\Pi_{u_\sigma} := \boldsymbol{\sigma} \cdot \boldsymbol{n} - \boldsymbol{\sigma}_D \ \text{ on } \ \partial\Omega_\sigma \times \mathbb{T}, \tag{16}$$

and for the mass balance equation Eq (10), we define the $\Pi_p$ as:

$$\Pi_p := \left( \phi c_f + \frac{\alpha - \phi}{K_s} \right) \frac{\partial p}{\partial t} + \alpha \frac{\partial \nabla \cdot \boldsymbol{u}}{\partial t} - \nabla \cdot \mathcal{N}[\boldsymbol{\kappa}](\nabla p - \rho \mathbf{g}) - g \ \text{ in } \ \Omega \times \mathbb{T}, \tag{17}$$

and for its $\partial\Omega_q$ according to Eq (12)

$$\Pi_{p_q} := -\mathcal{N}[\boldsymbol{\kappa}](\nabla p - \rho \mathbf{g}) \cdot \boldsymbol{n} - q_D \ \text{ on } \ \partial\Omega_q \times \mathbb{T}. \tag{18}$$

Demanding the $\Pi$ terms above to be as close to zero as possible corresponds to fulfilling Eqs (6), (8), (10) and (12).

**Loss function definition.**   The loss function applied with the PINN scheme is composed of two parts (here we use a mean squared error—$MSE$ as the metric). The error on the training data ($MSE_{tr}$) and the mean square value of the regularization term given by the physics-informed function ($MSE_\Pi$):

$$MSE = MSE_{tr} + MSE_\Pi, \tag{19}$$

where

$$MSE_\Pi = MSE_{\Pi_\Omega} + MSE_{\Pi_{\partial\Omega_N}}, \tag{20}$$

$$MSE_{\Pi_\Omega} = MSE_{\Pi_u} + MSE_{\Pi_p}, \tag{21}$$

and

$$MSE_{\Pi_{\partial\Omega_N}} = MSE_{\Pi_{u_\sigma}} + MSE_{\Pi_{p_q}}. \tag{22}$$

where $MSE_{\Pi_u}$, $MSE_{\Pi_{u_\sigma}}$, $MSE_{\Pi_p}$, and $MSE_{\Pi_{p_q}}$ correspond to the loss function of Eqs (15), (16), (17) and (18), respectively. The Dirichlet boundary conditions given on $\partial\Omega_D$ with respect to the linear momentum Eq (7) and mass balance Eq (11) equations are automatically incorporated into the $MSE_{tr}$.

A graphical presentation of how boundary points, initial points, as well as domain data, are used for training the neural networks is provided in Fig 2. For the forward model, the set of points that constitutes $\partial\Omega_D \times \mathbb{T}$ and $\Omega$ at $t = 0$ contributes to the $MSE_{tr}$ term of the loss

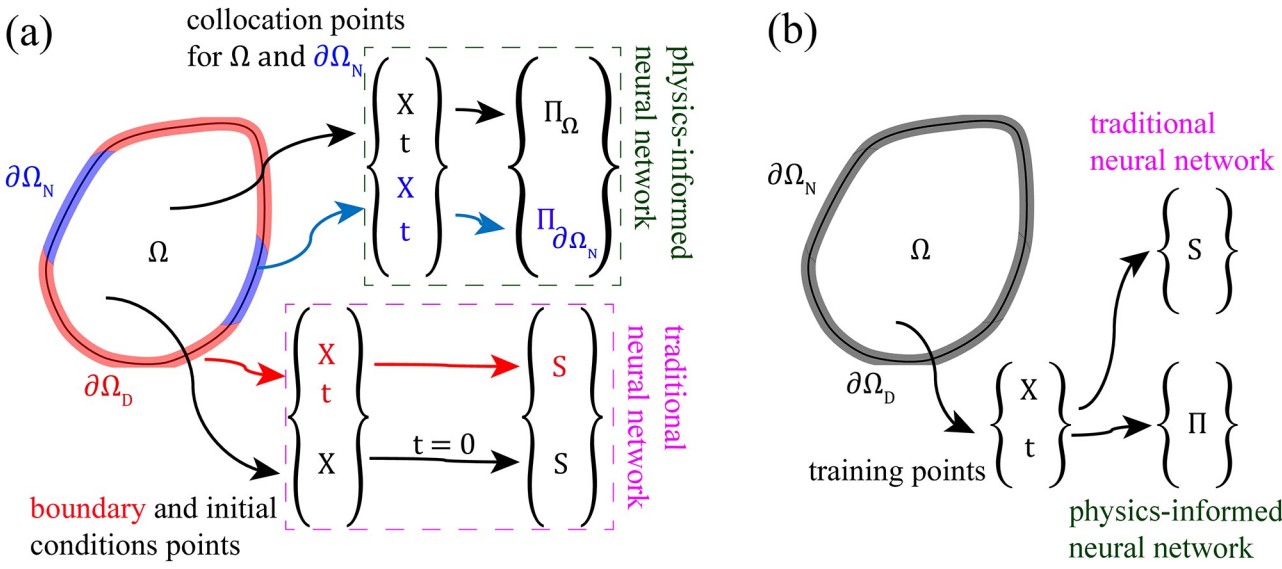

**Fig 2. Illustration of the parts of input space used for training the PINN.** (a) forward model and (b) inverse model. The collocation ($\Omega \times \mathbb{T}$), boundary and initial points ($\partial\Omega_D \times \mathbb{T}$ and $\Omega$ at $t = 0$) are utilized in the forward model. However, only the training points ($\Omega \times \mathbb{T}$) are employed in the inverse model. X represents a set of spatial coordinates ($x$, $y$, and $z$), $t$ is a set of coordinates in the time domain, and S is a set of solution values ($\boldsymbol{u}$ and $p$) corresponding to X and $t$.

function. We denote this set of training data as $N_b$. Besides, we have a set of collocation points that are sampled from $\partial\Omega_N \times \mathbb{T}$ and $\Omega \times \mathbb{T}$. These training data are used to minimize the $\Pi$ parts of the loss function, and we denote these data points as $N_\Pi$.

For the inverse model, we will have a set of known/measured data points that can be sampled from the whole domain, i.e., $\Omega \times \mathbb{T}$. We refer to this set of training data as $N_{tr}$. All this data will contribute to both the $MSE_{tr}$ and the $MSE_{\Pi_\Omega}$ terms of the loss function. Also, we may include an extra set of collocation points (i.e., data where we would have no measured values), which would contribute only to the $MSE_{\Pi_\Omega}$ term.

**Training the PINN.** The structure of a PINN architecture is shown in Fig 3. We assume a simple case of two inputs, $I_1$ and $I_2$, one output, O, and one hidden layer with two neurons. Moreover, we here assume that $\Pi$ is a function of O, the first derivative of O with respect to the set of inputs, and the set of physical parameters, $\theta$. With $(\cdot)_{tr}$ we represent a training set, where the O values are known for given $I_{1,tr}$ and $I_{2,tr}$. This set is used to evaluate the $MSE_{tr}$ term of Eq (19) at each training cycle. The $MSE_\Pi$ part of Eq (19) is obtained by calculating $\Pi$ for the collocation points $(\cdot)_\Pi$; see bottom part of Fig 3. Calculating $\Pi$ involves knowing O and its derivatives with respect to $I_1$ and $I_2$. As these values are unknown for the collocation points, they are estimated using the current approximation of $O(W, b, I_1, I_2)$ and its derivatives, which can be obtained using automatic differentiation [47, 48]. In this way, the regularization term $MSE_\Pi$ indirectly depends on the weights, W, and biases, b, of the neural architecture shown at the top of Fig 3. As a result, $\Pi$ can be written as a function of W, b, $I_{1,\Pi}$, $I_{2,\Pi}$, and $\theta$. If we denote this function, $\gamma$, we have $\gamma(W, b, I_{1,\Pi}, I_{2,\Pi}, \theta) = \Pi\left(O(W, b, I_{1,\Pi}, I_{2,\Pi}), \frac{\partial O}{\partial I_1}, \frac{\partial O}{\partial I_2}, \theta\right)$. Note that the specific mapping of $I_{1,\Pi}$ and $I_{2,\Pi}$ to $\Pi$ has been defined as the physical informed neural network in previous work [26] as opposed to the neural network shown in the top of Fig 3.

For both the forward and the inverse modeling cases, one trains the neural networks to establish a mapping from the input space given by $I_1$ and $I_2$ to the output space, O, by minimizing $MSE_{tr}$ and $MSE_\Pi$. The essential differences between the two cases come down to the type

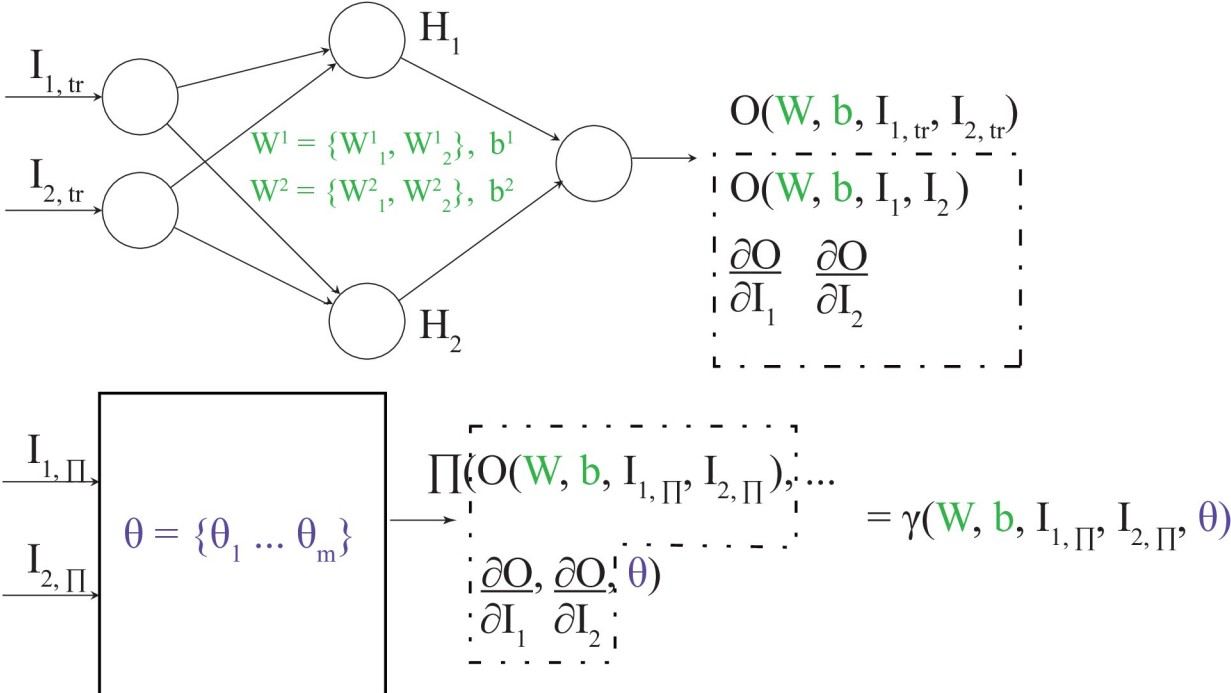

**Fig 3. Graphical illustration of how a traditional neural network is linked to a physics-informed neural network.** The set of $\theta$ represents unknown physical parameters that we want to estimate.

of training examples being available to train the networks, as discussed in Fig 2, and whether the physical parameters ($\theta$) are known or not. For the forward modeling, we apply $(\cdot)_{tr}$ to evaluate the $MSE_{tr}$ term and $(\cdot)_\Pi$ to evaluate the $MSE_\Pi$ term. The W and b are then adjusted to minimize the sum of these terms. One should emphasize that for the forward modeling, all the variables to be learned belong to the neural network depicted in the top of Fig 3.

For the inverse modeling, the aim is to estimate $\theta$. We still, however, train a neural network to predict O (similar to the forward case). That is, we are not using a loss function that involves measuring a distance between estimated values of $\theta$ and their ground truth values. Instead, the reasoning behind solving the inverse problem is that we expect the unknown $\theta$ to converge towards their true values during training because we also allow $\theta$ to be adjusted along with W and b. During a training phase, these variables are learned to minimize the combined sum of $MSE_{tr}$ and $MSE_\Pi$. Specifically, the variables are adjusted by backpropagating the errors as calculated by Eq (19) [44, 45]. Unlike the forward problem, the boundary and initial conditions are unknown and cannot be used to generate training examples. Instead, we provide training examples that, in real cases, would be obtained from measurements, which ideally correspond to solution points of the forward problem inside $\Omega$ (see Fig 2b).

As discussed above, the solution of the inverse problem is based on training the neural network to establish a mapping from $I_1$ and $I_2$ to O as we do in the forward case. This means that the hyperparameters estimated from the forward modeling may act as qualified estimates for the hyperparameters in the inverse case, assuming the number of training examples in both cases are similar. By definition of the inverse problem, we do not know the true values of $\theta$, so we would have to assume that the exact values of $\theta$ have a limited influence on the hyperparameters. A more direct way of estimating the hyperparameters in the inverse case would be to divide the data into training and validation sets and then select the hyperparameters that

minimize *MSE* with respect to learning the mapping from $I_1$ and $I_2$ to $O$. The downside of this is that we could then not spend all the measurement data on training the neural networks.

## Results and discussion

We first consider the results of forward and inverse models of the nonlinear diffusivity equation. Subsequently, we present the results of the nonlinear Biot's equations for both the forward and inverse models.

### Nonlinear diffusivity equation

We assume $\alpha = 0$ to decouple the momentum and mass balance equations and focus on Eq (10), which is reduced to

$$\phi c_t \frac{\partial p}{\partial t} - \nabla \cdot \mathcal{N}[\boldsymbol{\kappa}](\nabla p - \rho \mathbf{g}) = g \text{ in } \Omega \times \mathbb{T}, \tag{23}$$

where $c_t$ is a total compressibility. The same boundary and initial conditions, Eqs (11) to (13), are still valid.

We take $\Omega = [0, 1]^1$, $\mathbb{T} = [0, 1]$, and choose the exact solution in $\Omega$ as:

$$p(x, t) := \sin(x + t), \tag{24}$$

and $\mathcal{N}[\boldsymbol{\kappa}]$ as:

$$\mathcal{N}[\boldsymbol{\kappa}] := \boldsymbol{\kappa_0} p^2, \tag{25}$$

where $x$ and $t$ represent points in x-direction and time domain, respectively. The $\boldsymbol{\kappa_0}$ is assumed to be a scalar in this case, i.e., $\boldsymbol{\kappa_0} = \kappa_0$. All the physical constants are set to 1.0; and subsequently, $g$ is chosen as:

$$g(x, t) := \cos(x + t) + \sin(x + t)^3 - 2\cos(x + t)^2 \sin(x + t), \tag{26}$$

to satisfy the exact solution. Furthermore, the homogeneous boundary conditions are applied to all boundaries and initial conditions using Eq (24). Combining Eqs (23) to (26), the physics-informed function ($\Pi$) for the nonlinear diffusivity equation is defined:

$$\Pi(x, t) := \phi c_t \frac{\partial p}{\partial t} - \kappa_0 \frac{\partial}{\partial x} p^2 \left( \frac{\partial}{\partial x} p \right) - g. \tag{27}$$

We generate the solution based on an interval mesh having 2559 equidistant spatial intervals and 99 temporal ones; hence, in total, we have 256000 solution points, including the points on boundaries. Subsequently, we randomly draw $n$ training examples. Half of the remaining points are randomly selected as a validation set, and the remaining ones are used for testing. As an illustration, if we have 256000 solution points and use 100 examples to train the model. We then use 127950 examples for the validation and 127950 examples for the test set.

For the nonlinear diffusivity equation, the forward modeling with a neural network should calculate the $p$ at any given $x$ and $t$ from provided values of $\phi$, $c_t$, and $\kappa_0$. For the inverse case, the aim is to infer $\phi$, $c_t$, and $\kappa_0$ from observed values of $x$, $t$, and $p$. The architecture of the neural network corresponding to the top of Fig 3 is illustrated in Fig 4. We have two input nodes and one output node for this case. We use L-BFGS [49]; a quasi-Newton, fullbatch gradient-based optimization algorithm to minimize the loss function with stop criteria as $\frac{|MSE^k - MSE^{k+1}|}{\max(|MSE^k|, |MSE^{k+1}|, 1.0)} <= 10^{-16}$ where $(\cdot)^k$ and $(\cdot)^{k+1}$ are previous and current iteration, respectively. The L-BFGS algorithm has several advantages that are suitable for this study; for

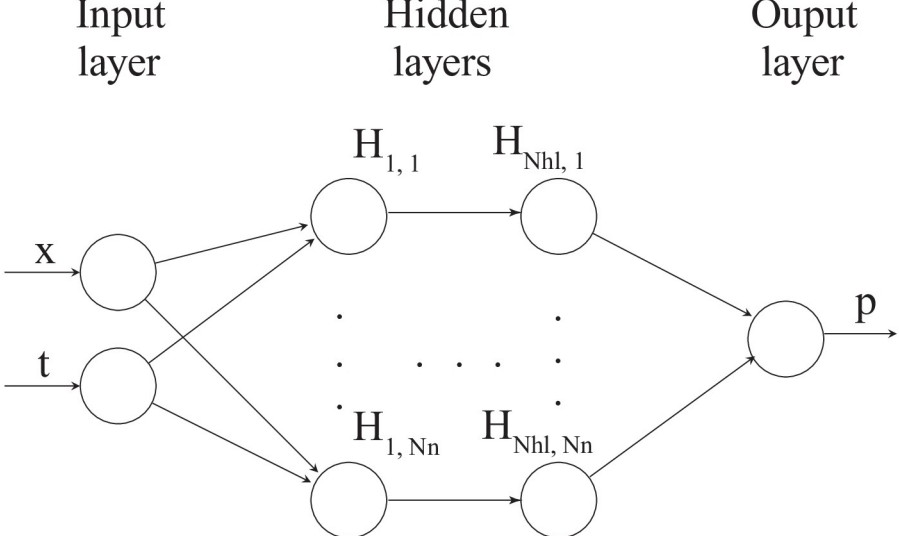

**Fig 4. Neural network architecture used for nonlinear diffusivity equation.** This network corresponds to the top part of Fig 3. There are two input nodes, $x$ and $t$, and one output node, $p$. The number of hidden layers, $N_{hl}$, and the number of neurons for each hidden layer, $N_n$, denote the hyperparameters.

example, it is more stable compared to stochastic gradient descent (SGD) and can handle large batch sizes well [49, 50]. This setting is used for all the simulations presented in this paper unless stated otherwise.

To reiterate, the sampling strategies of the forward and inverse modelings are illustrated in Fig 2. To be specific, in the forward case, we must determine the solution of the partial differential equation based on known boundary values for the time interval of $\mathbb{T} = (0, \tau]$ combined with the initial values at $t = 0$. These boundary and initial points are used to calculate the $MSE_{tr}$ term of Eq 19. The inner points act as collocation points and are used to calculate the $MSE_{\Pi}$ term of Eq 19. Note that for the forward case, we can pick as many points as we wish to calculate for both terms of the loss function. For the inverse model, we know a set of $x$ and $t$ and their corresponding values of $p$ in advance. These examples are used to calculate the $MSE_{tr}$ and $MSE_{\Pi}$ terms of Eq 19. In a real setting, the amount of training examples is limited by the available measurements.

**Verification of the forward model.** Following Eq (19) we obtain:

$$MSE = MSE_b + MSE_{\Pi}, \tag{28}$$

where

$$MSE_b = \frac{1}{N_b} \sum_{i=1}^{N_b} \left| p\left(x_b^i, t_b^i\right) - p^i \right|^2 \tag{29}$$

and

$$MSE_{\Pi} = \frac{1}{N_{\Pi}} \sum_{i=1}^{N_{\Pi}} \left| \Pi\left(x_{\Pi}^i, t_{\Pi}^i\right) \right|^2, \tag{30}$$

where $\{x_b^i, t_b^i, p^i\}_{i=1}^{N_b}$ refer to the initial and boundary training data on $p(x, t)$ while $\{x_{\Pi}^i, t_{\Pi}^i\}_{i=1}^{N_{\Pi}}$ denote the collocation points for $\Pi$, which are sampled using the Latin Hypercube method

**Table 1. Diffusivity equation—forward modeling: PINN performance as function of training set size.** Relative $\mathcal{L}^2$ errors between the exact and predicted values of $p$ for the validation set. The table shows the dependency on the number of the initial and boundary training data, $N_b$, and on the number of collocation points, $N_\Pi$. Here, the network architecture is fixed to 4 layers with 10 neurons per hidden layer.

Panel A: Average over 1 realization

| $N_\Pi$ | 2 | 5 | 10 | 20 | 40 | 80 | 160 | 320 | 640 |
|---|---|---|---|---|---|---|---|---|---|
| $N_b$ | | | | | | | | | |
| 2 | 0.1010 | 0.1150 | 0.4060 | 0.2570 | 0.1530 | 0.0108 | 0.1930 | 0.0236 | 0.0216 |
| 3 | 0.0339 | 0.0665 | 0.1050 | 0.0122 | 0.0020 | 0.0973 | 0.0025 | 0.0314 | 0.0074 |
| 6 | 0.0248 | 0.0401 | 0.1110 | 0.0807 | 0.0012 | 0.0057 | 0.0014 | 0.0116 | 0.0046 |
| 12 | 0.0072 | 0.1230 | 0.0126 | 0.0296 | 0.0012 | 0.0119 | 0.0010 | 0.0015 | 0.0076 |
| 24 | 0.0152 | 0.0018 | 0.0008 | 0.0008 | 0.0006 | 0.0016 | 0.0008 | 0.0006 | 0.0009 |
| 48 | 0.0124 | 0.0127 | 0.0015 | 0.0037 | 0.0044 | 0.0013 | 0.0035 | 0.0002 | 0.0026 |
| 96 | 0.0076 | 0.0010 | 0.0023 | 0.0015 | 0.0026 | 0.0017 | 0.0009 | 0.0007 | 0.0010 |
| 192 | 0.0036 | 0.0006 | 0.0205 | 0.0025 | 0.0010 | 0.0051 | 0.0005 | 0.0008 | 0.0002 |
| 384 | 0.0020 | 0.0019 | 0.0009 | 0.0004 | 0.0008 | 0.0001 | 0.0005 | 0.0005 | 0.0003 |

Panel B: Average over 3 realizations

| $N_\Pi$ | 2 | 5 | 10 | 20 | 40 | 80 | 160 | 320 | 640 |
|---|---|---|---|---|---|---|---|---|---|
| $N_b$ | | | | | | | | | |
| 2 | 0.0972 | 0.0596 | 0.0417 | 0.0859 | 0.0380 | 0.0086 | 0.0041 | 0.0105 | 0.0050 |
| 3 | 0.0744 | 0.0652 | 0.0969 | 0.0415 | 0.0173 | 0.0268 | 0.0261 | 0.0084 | 0.0077 |
| 6 | 0.1020 | 0.1160 | 0.0841 | 0.0133 | 0.0071 | 0.0756 | 0.0217 | 0.0078 | 0.0064 |
| 12 | 0.0853 | 0.0099 | 0.0751 | 0.0088 | 0.0061 | 0.0456 | 0.0036 | 0.0059 | 0.0033 |
| 24 | 0.0447 | 0.0272 | 0.0104 | 0.0214 | 0.0290 | 0.0015 | 0.0041 | 0.0017 | 0.0060 |
| 48 | 0.0201 | 0.0037 | 0.0012 | 0.0230 | 0.0040 | 0.0072 | 0.0042 | 0.0060 | 0.0025 |
| 96 | 0.0104 | 0.0020 | 0.0013 | 0.0024 | 0.0048 | 0.0011 | 0.0009 | 0.0019 | 0.0020 |
| 192 | 0.0186 | 0.0014 | 0.0022 | 0.0006 | 0.0007 | 0.0011 | 0.0006 | 0.0004 | 0.0008 |
| 384 | 0.0015 | 0.0011 | 0.0010 | 0.0009 | 0.0005 | 0.0007 | 0.0010 | 0.0005 | 0.0005 |

Panel C: Average over 24 realizations

| $N_\Pi$ | 2 | 5 | 10 | 20 | 40 | 80 | 160 | 320 | 640 |
|---|---|---|---|---|---|---|---|---|---|
| $N_b$ | | | | | | | | | |
| 2 | 0.1002 | 0.0965 | 0.0904 | 0.0732 | 0.0849 | 0.0514 | 0.0139 | 0.0194 | 0.0188 |
| 3 | 0.0879 | 0.0775 | 0.0861 | 0.0513 | 0.0352 | 0.0354 | 0.0260 | 0.0111 | 0.0117 |
| 6 | 0.0710 | 0.0537 | 0.0529 | 0.0241 | 0.0144 | 0.0265 | 0.0153 | 0.0056 | 0.0107 |
| 12 | 0.0541 | 0.0351 | 0.0376 | 0.0302 | 0.0104 | 0.0235 | 0.0111 | 0.0135 | 0.0058 |
| 24 | 0.0320 | 0.0233 | 0.0141 | 0.0311 | 0.0160 | 0.0052 | 0.0061 | 0.0043 | 0.0039 |
| 48 | 0.0178 | 0.0050 | 0.0162 | 0.0111 | 0.0040 | 0.0024 | 0.0044 | 0.0041 | 0.0026 |
| 96 | 0.0087 | 0.0037 | 0.0031 | 0.0034 | 0.0022 | 0.0015 | 0.0021 | 0.0010 | 0.0012 |
| 192 | 0.0064 | 0.0049 | 0.0017 | 0.0010 | 0.0010 | 0.0011 | 0.0012 | 0.0009 | 0.0009 |
| 384 | 0.0046 | 0.0026 | 0.0011 | 0.0011 | 0.0010 | 0.0007 | 0.0008 | 0.0007 | 0.0006 |

provided within the pyDOE package [51]. The $N_b$ is the combined number of initial and boundary training data, and $N_\Pi$ is the number of collocation points.

In Table 1, we explore the accuracy of the PINN as a function of the number of training examples, i.e., different numbers of $N_b$ and $N_\Pi$ for a given size of the network; $N_{hl}$ and $N_n$ are fixed to four and ten, respectively. Due to the stochastic behavior of the training procedure of the neural networks, we calculate the results as an average over many realizations as it otherwise might be difficult to observe a clear trend of the relative $\mathcal{L}^2$ error as a function of the number of training examples (see Panel A of Table 1). From Panels B and C of Table 1, we observe

**Table 2. Diffusivity equation—forward modeling: PINN performance as function of hyperparameters.** Relative $\mathcal{L}^2$ errors between the exact and predicted values of $p$ for the validation set. The table shows the dependency on the different number of hidden layers $N_{hl}$ and different number of neurons per layer $N_n$. Here, the total number of training and collocation points is fixed to $N_b = 96$ and $N_\Pi = 160$, respectively.

| Panel A: Average over 1 realization | | | | | | |
|---|---|---|---|---|---|---|
| $N_n$ | 2 | 5 | 10 | 20 | 40 | 80 |
| $N_{hl}$ | | | | | | |
| 2 | 0.0009 | 0.0015 | 0.0062 | 0.0003 | 0.0003 | 0.0088 |
| 4 | 0.0033 | 0.0036 | 0.0098 | 0.0009 | 0.0144 | 0.0007 |
| 6 | 0.0003 | 0.0038 | 0.0017 | 0.0175 | 0.0268 | 0.0009 |
| 8 | 0.0021 | 0.0010 | 0.0005 | 0.0005 | 0.0006 | 0.0058 |
| 16 | 0.0411 | 0.0003 | 0.0105 | 0.0007 | 0.0175 | 0.0019 |
| 32 | 0.0735 | 0.0019 | 0.0073 | 0.1970 | 0.0024 | 0.0081 |
| Panel B: Average over 3 realizations | | | | | | |
| $N_n$ | 2 | 5 | 10 | 20 | 40 | 80 |
| $N_{hl}$ | | | | | | |
| 2 | 0.0027 | 0.0015 | 0.0021 | 0.0042 | 0.0025 | 0.0026 |
| 4 | 0.0004 | 0.0025 | 0.0013 | 0.0024 | 0.0014 | 0.0025 |
| 6 | 0.0023 | 0.0021 | 0.0020 | 0.0028 | 0.0026 | 0.0025 |
| 8 | 0.0049 | 0.0010 | 0.0023 | 0.0017 | 0.0016 | 0.0020 |
| 16 | 0.5580 | 0.0029 | 0.0012 | 0.0030 | 0.0019 | 0.0027 |
| 32 | 0.1150 | 0.0328 | 0.0036 | 0.0034 | 0.0107 | 0.0022 |
| Panel C: Average over 24 realizations | | | | | | |
| $N_n$ | 2 | 5 | 10 | 20 | 40 | 80 |
| $N_{hl}$ | | | | | | |
| 2 | 0.1660 | 0.0022 | 0.0032 | 0.0025 | 0.0025 | 0.0020 |
| 4 | 0.0171 | 0.0024 | 0.0018 | 0.0021 | 0.0018 | 0.0022 |
| 6 | 0.0032 | 0.0014 | 0.0018 | 0.0024 | 0.0020 | 0.0023 |
| 8 | 0.0253 | 0.0016 | 0.0019 | 0.0020 | 0.0021 | 0.0025 |
| 16 | 0.0912 | 0.0019 | 0.0019 | 0.0019 | 0.0025 | 0.0033 |
| 32 | 0.0462 | 0.0090 | 0.0037 | 0.0027 | 0.0036 | 0.0031 |

that the average accuracy of the PINN is improved as $N_b$ and $N_\Pi$ are increased. Moreover, the combined number of $N_b$ and $N_\Pi$ to obtain an average value of the $\mathcal{L}^2$ error below $1.0 \times 10^{-2}$ is in the order of 100s. To illustrate the impact of $\Pi$ terms, we consider the case where we simply train a neural network to interpolate a feed-forward solution based on a known set of solution points. Without $\Pi$ regularization we obtain an average relative $\mathcal{L}^2$ error of $5.6 \times 10^{-2}$ based on 96 training examples while the average $\mathcal{L}^2$ error becomes less than $1.0 \times 10^{-2}$ when including $\Pi$.

Next, we perform a sensitivity analysis to explore how the PINN performance depends on the hyperparameters, $N_{hl}$ and $N_n$. We do this using a fixed size of the training set with $N_b$ and $N_\Pi$ equal to 96 and 160, respectively. Ideally, the explored space of the hyperparameters should reveal a parameter set that produces a minimum of the loss function on a validation set. To deal with the stochastic nature of the training procedure, again, we calculate average values obtained over many realizations. The results are presented in Table 2. From panel C, we observe that selecting $N_{hl} = 6$ and $N_n = 5$ corresponds to the best accuracy in the explored space. For a smaller and larger size of the architecture, the accuracy decreases, corresponding to underfitting and overfitting, respectively [52]. We now apply the trained PINN model using

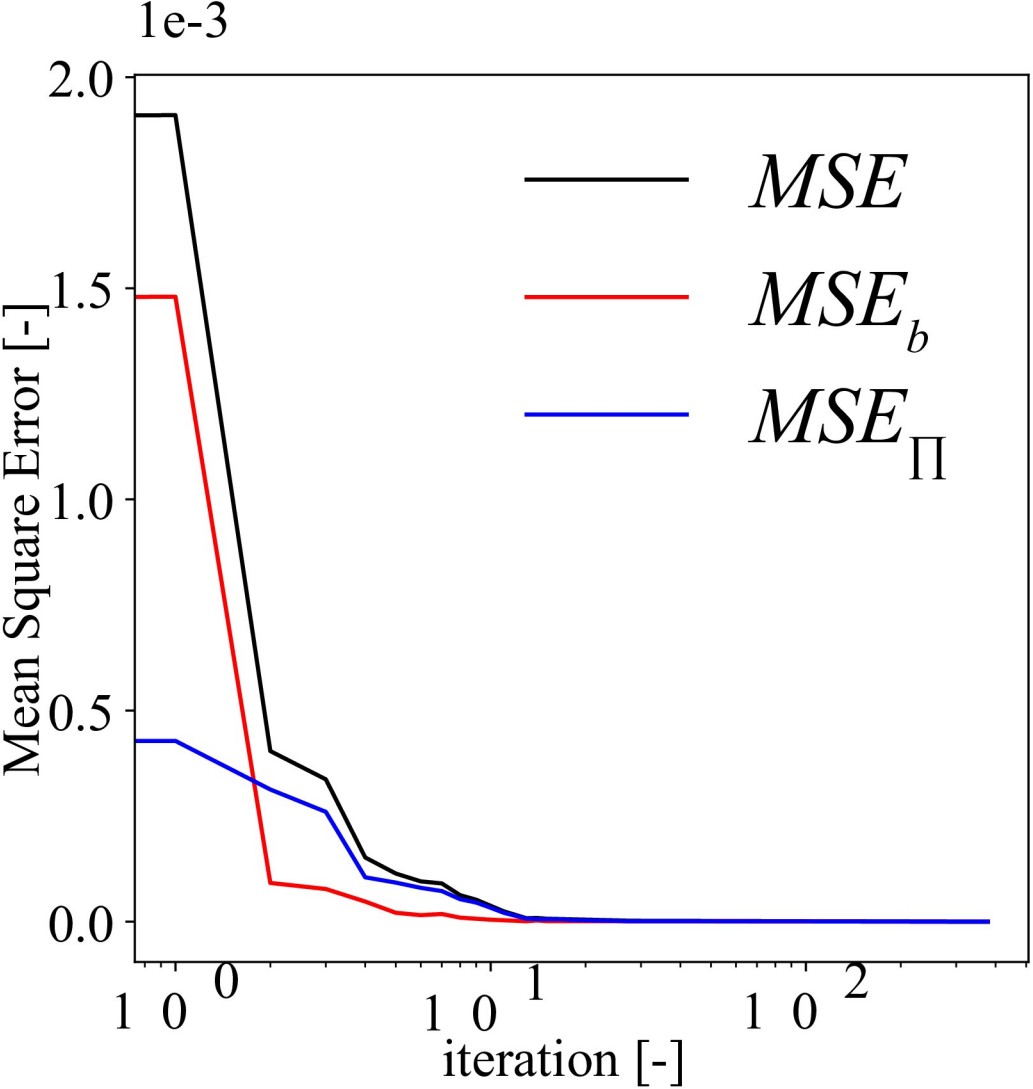

**Fig 5. Diffusivity equation—forward modeling: mean square training error plot.** This model uses $N_{hl} = 6$, $N_n = 5$, $N_b = 96$, and $N_\Pi = 160$. *MSE*, *MSE*$_b$, and *MSE*$_\Pi$ are calculated using Eqs 28, 29 and 30, respectively.

$N_b = 96$, $N_\Pi = 160$, $N_{hl} = 6$, and $N_n = 5$ to the test set. The relative $\mathcal{L}^2$ error is $1.43 \times 10^{-3}$, which is comparable to that of the validation set (see Table 2—Panel C).

For the selected model ($N_{hl} = 6$, $N_n = 5$, $N_b = 96$, and $N_\Pi = 160$), we illustrate the behavior of the mean square training error of this model as function of the training iterations in Fig 5. One can observe that $MSE_b$ is higher than $MSE_\Pi$ in the beginning, but later it approaches zero faster than $MSE_\Pi$. The *MSE* converges steadily without any oscillations.

**Inverse model of diffusivity equation.** We rewrite Eq (27) in the parametrized form [23] as presented below:

$$\Pi(x, t) := \theta_1 \frac{\partial p}{\partial t} - \theta_2 \frac{\partial}{\partial x} p^2 \left( \frac{\partial}{\partial x} p \right) - g, \tag{31}$$

where

$$\theta_1 = \phi c_t \quad \text{and} \quad \theta_2 = \kappa_0. \tag{32}$$

Unlike the forward mode, where the weights and biases are the unknown parameters to be learned, we now have two more unknowns, $\theta_1$ and $\theta_2$. Using Eq (19) we have:

$$MSE_{tr} = \frac{1}{N_{tr}} \sum_{i=1}^{N_{tr}} |p(x_{tr}^i t_{tr}^i) - p^i|^2 \tag{33}$$

and

$$MSE_\Pi = \frac{1}{N_{tr}} \sum_{i=1}^{N_{tr}} |\Pi(x_{tr}^i, t_{tr}^i)|^2, \tag{34}$$

where $\{x_{tr}^i, t_{tr}^i, p^i\}_{i=1}^{N_{tr}}$ refers to the set of training data and $N_{tr}$ is the number of training data. In contrast to the forward model, we do not need to specify specific collocation points to activate the $\Pi$ dependent loss term; here we can just use the training examples used to calculate $MSE_{tr}$. All physical constants are set to one; hence, $\theta_1 = 1.0$ and $\theta_2 = 1.0$. Note that $\theta_1$ and $\theta_2$ are considered constant throughout the domain.

We use the hyperparameters obtained from the sensitivity analysis of the forward model (see Panel C of Table 2), i.e., $N_{hl} = 6$ and $N_n = 5$. In Table 3, we illustrate that this choice also yields the least $\mathcal{L}^2$ error of $p$ and percentage errors of $\theta_1$ and $\theta_2$ for the inverse problem, supporting the heuristic arguments presented above in the section "Training the PINN". Moreover, we find that the $\mathcal{L}^2$ error of $p$ and percentage error of $\theta_1$ and $\theta_2$ are not much different with different combinations of the hyperparameters. Note that the $\mathcal{L}^2$ error of $p$ and percentage errors of $\theta_1$ and $\theta_2$ presented in Table 3 represent average values over 24 realizations.

We depict the performance of the PINN model for solving the inverse problem as a function of $N_{tr}$ in Fig 6. The reported error values of $\theta_1$ and $\theta_2$ are percentage errors, while the relative $\mathcal{L}^2$ error is shown for $p$. The error bars show the standard derivation (±1 SD) based on 24 realizations. We observe that a minimum of $N_{tr} = 200$ is required by the PINN model to avoid substantial stochastic fluctuations in the estimated values of $\theta_1$ and $\theta_2$. Moreover, we observe that the PINN model with $N_{tr} > 200$ provides average percentage errors of $\theta_1$ and $\theta_2$ less than

**Table 3. Diffusivity equation—inverse modeling: PINN performance as function of hyperparameters.** Relative $\mathcal{L}^2$ error of $p$ and percentage error of $\theta_1$ and $\theta_2$ for different number of hidden layers, $N_{hl}$, and different number of neurons per layer, $N_n$. The $N_{tr}$ is fixed at 250. Note that we pick the optimal hyperparameters, i.e., $N_{hl} = 6$ and $N_n = 5$ from the sensitivity analysis of the forward model. Results shown in this table are an average over 24 realizations.

| | $N_n$ | 2 | 5 | 10 |
|---|---|---|---|---|
| | $N_{hl}$ | | | |
| $p$ | 4 | 0.0004 | 0.0003 | 0.0003 |
| | 6 | 0.0007 | 0.0002 | 0.0004 |
| | 8 | 0.0010 | 0.0002 | 0.0003 |
| $\theta_1$ | 4 | 0.1888 | 0.1357 | 0.1969 |
| | 6 | 0.3200 | 0.1065 | 0.3261 |
| | 8 | 0.5195 | 0.1272 | 0.1125 |
| $\theta_2$ | 4 | 0.3443 | 0.3562 | 0.3289 |
| | 6 | 0.7121 | 0.1912 | 0.6946 |
| | 8 | 0.7949 | 0.3088 | 0.2504 |

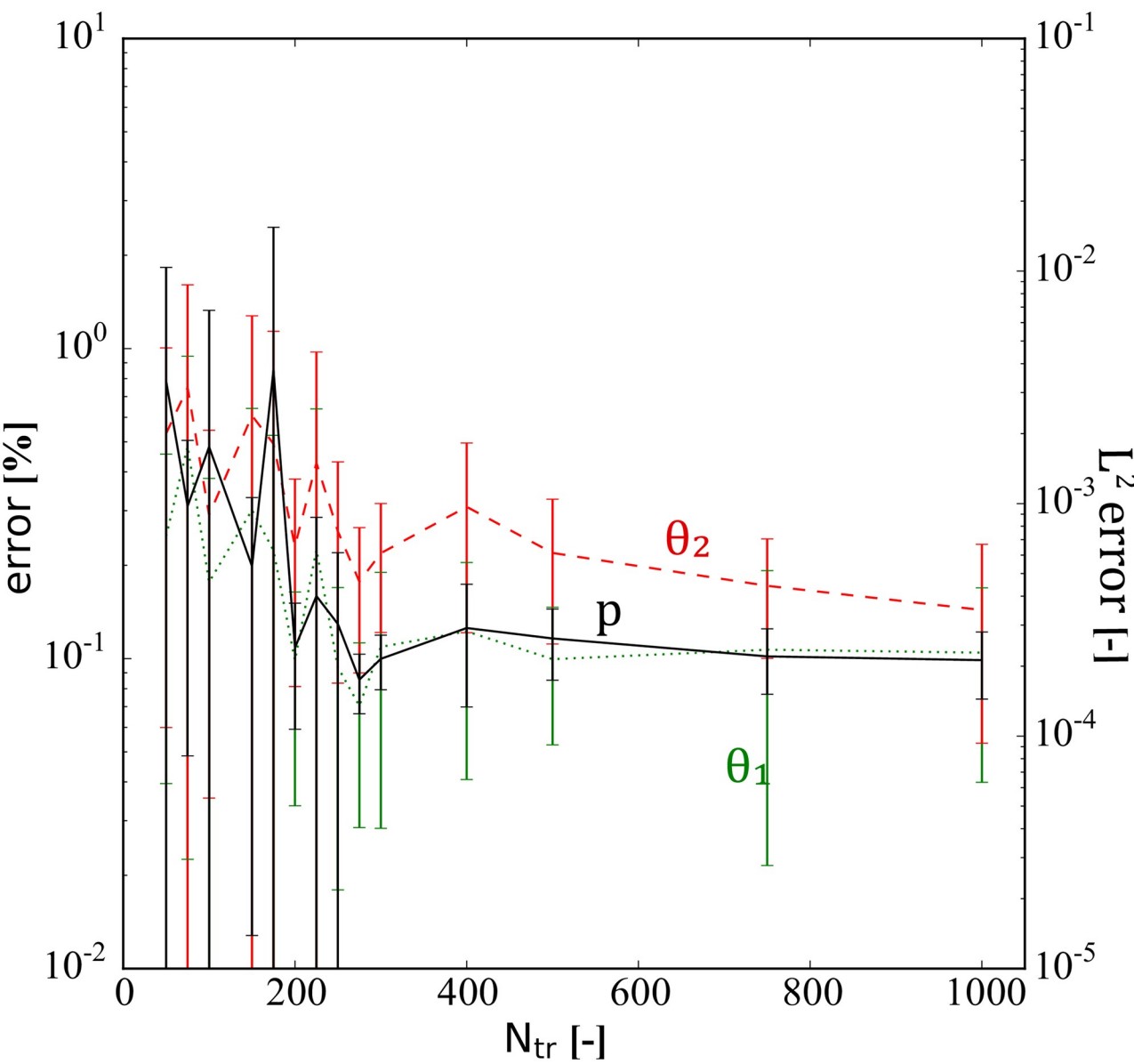

**Fig 6. Diffusivity equation—inverse modeling: PINN performance as function of training set size.** This figure shows the estimated error dependency on the amount of training data, $N_{tr}$. The error bars show mean and standard derivation (± 1 SD) based on 24 realizations. Note that there is no noise added in this investigation, and all physical constants are set to one, i.e., $\theta_1 = \theta_2 = 1.0$. The reported error values of $\theta_1$ and $\theta_2$ are percentage errors while the relative $\mathcal{L}^2$ error is shown for $p$.

1%, but with a large standard deviation. Over 24 realizations of the trained PINN with $N_{tr} = 200$, the minimum percentage errors are $2.31 \times 10^{-2}$ and $7.33 \times 10^{-2}$ and the maximum are $1.93 \times 10^{-1}$ and $4.92 \times 10^{-1}$, for $\theta_1$ and $\theta_2$, respectively. To obtain 200 training examples in a real setting, i.e., lab experiments or field observations, is realistic; hence, the results illustrate the feasibility of PINN to solve the inverse problem based on a reasonably sized data set.

Next, we perform a systematic study of the effect of additive noise in data, which is created from the true data as follows [26]:

$$\boldsymbol{X}_{noise} = \boldsymbol{X}_{true} + \epsilon \mathcal{S}(\boldsymbol{X}_{true})\mathcal{G}(0,1), \tag{35}$$

**Table 4. Diffusivity equation—inverse modeling: PINN performance as function of noise.** This figure shows the average percentage errors of $\theta_1$ and $\theta_2$ for different numbers of training data, $N_{tr}$, as function of the noise levels. Here, the neural network architecture is kept fixed to 6 layers and 5 neurons per layer. The results are averages over 10 realizations.

| Noise ($\epsilon$) | | 0% | 1% | 5% | 10% |
|---|---|---|---|---|---|
| $N_{tr}$ | | | | | |
| $\theta_1$ | 100 | 0.17 | 1.82 | 4.40 | 4.67 |
| | 250 | 0.15 | 0.30 | 1.00 | 1.98 |
| | 500 | 0.12 | 0.17 | 0.77 | 0.84 |
| | 1000 | 0.04 | 0.09 | 0.34 | 0.94 |
| $\theta_2$ | 100 | 0.22 | 1.49 | 4.35 | 4.90 |
| | 250 | 0.24 | 0.52 | 1.80 | 2.45 |
| | 500 | 0.23 | 0.47 | 0.99 | 1.48 |
| | 1000 | 0.13 | 0.28 | 0.41 | 0.79 |

where $X_{noise}$ and $X_{true}$ is the vector of the data with and without noise, respectively. The $\epsilon$ determines the noise level, $\mathcal{S}(\cdot)$ represents a standard deviation operator, $\mathcal{G}(0, 1)$ is a random value, which is sampled from the Gaussian distribution with mean and standard deviation of zero and one, respectively. The noise generated from this procedure is fully random and uncorrelated. The results are presented in Table 4 as average values over 10 training realizations. We can observe that the error increases with the noise level ($\epsilon$) while the error decreases as the $N_{tr}$ is increased. As expected, the PINN model requires more data to accurately approximate the unknown physical parameters when the noise level is high.

## Nonlinear Biot's equations

From the nonlinear diffusivity equation section, we have shown that the PINN model can solve forward and inverse problems. We then progress to the multiphysics problem represented by the nonlinear Biot's equations. We take $\Omega = [0, 1]^2$, $\mathbb{T} = [0, 1]$, and choose the exact solution in $\Omega$ as:

$$\boldsymbol{u}(x, y, t) := \begin{bmatrix} u \\ v \end{bmatrix} = \begin{bmatrix} \sin(x + y + t) \\ \cos(x + y + t) \end{bmatrix}, \tag{36}$$

for the displacement variable where $u$ and $v$ are displacements in x- and y-direction, respectively. Note that as we focus on the 2-Dimensional domain; therefore, $\boldsymbol{u}(x, y, t)$ is composed of two spatial components. For the pressure variable, we choose

$$p(x, y, t) := e^{(x+y+t)}. \tag{37}$$

Here $x$, $y$, and $t$ represent points in x-, y-direction, and time domain, respectively. The $\mathcal{N}[\boldsymbol{\kappa}]$ is chosen as:

$$\mathcal{N}[\boldsymbol{\kappa}] := \boldsymbol{\kappa_0} e^{\varepsilon_v}, \tag{38}$$

where $\boldsymbol{\kappa_0}$ represent initial rock matrix conductivity. Again, we assume $\boldsymbol{\kappa_0}$ to be a scalar in this case, i.e., $\boldsymbol{\kappa_0} = \kappa_0$. The $\varepsilon_v$ is the total volumetric strain defined as:

$$\varepsilon_v := \text{tr}(\boldsymbol{\varepsilon}) = \sum_{i=1}^{2} \varepsilon_{ii}. \tag{39}$$

The choice of $\mathcal{N}[\boldsymbol{\kappa}]$ function is selected to represent the change in a volumetric strain that

affects the porous media conductivity, and it is adapted from [53–55]. All the physical constants are set to 1.0; and subsequently, $f$ is chosen as:

$$\boldsymbol{f}(x, y, t) := \begin{bmatrix} f_u(x, y, t) \\ f_v(x, y, t) \end{bmatrix}, \tag{40}$$

where

$$f_u(x, y, t) := -4.0 \, \sin(x + y + t) - 2.0 \, \cos(x + y + t) - e^{(x+y+t)}, \tag{41}$$

and

$$f_v(x, y, t) := -4.0 \, \cos(x + y + t) - 2.0 \, \sin(x + y + t) - e^{(x+y+t)}, \tag{42}$$

for the momentum balance equation, Eq (6). The source term of the mass balance equation, Eq (10), $g$ is chosen as:

$$
\begin{aligned}
g(x, y, t) := {} & (\cos(x + y + t) + \sin(x + y + t) - 1)e^{\cos(x+y+t)-\sin(x+y+t)+x+y+t} \\
& -\cos(x + y + t) + e^{x+y+t} - \sin(x + y + t),
\end{aligned}
\tag{43}
$$

to satisfy the exact solution. Furthermore, the boundary conditions and initial conditions are applied using Eqs (36) and (37). The $\Pi_{\boldsymbol{u}}(x, y, t)$ and $\Pi_p(x, y, t)$, Eqs (15) and (17), here act as the physics-informed function.

We generate the exact solution points, Eqs (36) and (37), based on a rectangular mesh ($\Omega = [0, 1]^2$) with 99 equidistant intervals in both x- and y-direction, i.e. $\Delta x = \Delta y$. Using 49 equidistant temporal intervals, in total, we have 500000 examples. Similar to the diffusivity equation case, we draw $n$ training examples randomly. Subsequently, we split the remaining examples equally for validation and test sets. Again, assuming we have 500000 solution points for the sake of illustration, we use 100 examples to train the model; we then have 249950 examples for both the validation and the test sets.

To recap, the forward modeling of Biot's system aims to predict the displacement ($\boldsymbol{u}$) and pressure ($p$) by specifying the initial and boundary conditions, collocation points, and as well as the physical parameters ($\mu_l, \lambda_l, \alpha, \phi, c_f, K_s,$ and $\kappa_0$). The inverse modeling, however, aims to estimate the physical parameters from observed values of $\boldsymbol{u}$ and $p$ with their corresponding values of $x, y,$ and $t$.

In the case of the nonlinear Biot's equations, the architecture of the neural network corresponding to the top of Fig 3 is presented in Fig 7. We have three input nodes and three output nodes for this case. For the forward modeling, the hyperparameters $N_{hl}$ and $N_n$ are found using a sensitivity analysis, and as argued in the above section, "Training the PINN," we apply the same hyperparameters for the inverse model. Again, we use L-BFGS; a quasi-Newton, full-batch gradient-based optimization algorithm to minimize the loss function [49] for the forward model. For the inverse problem, we find that combining ADAM, stochastic gradient descent, and L-BFGS might provide faster convergence when training the neural network. Specifically, we use ADAM for the first 10000 iterations and then continue using L-BFGS until the stop criterion is met. Note that we apply the same stop criterion as described above for the diffusivity case. Since the ADAM is a first-order method compared to L-BFGS, which is a second-order model, ADAM is less computationally demanding. Initially, where the weights of the neural network are far from convergence, we speculate that the less computational effort of ADAM is an advantage. However, as we approach the minimum, L-BFGS is likely to provide a better estimate of the steepest descent. Whether these observations could be made when dealing with other types of partial differential equations is an open question, but the use of a similar

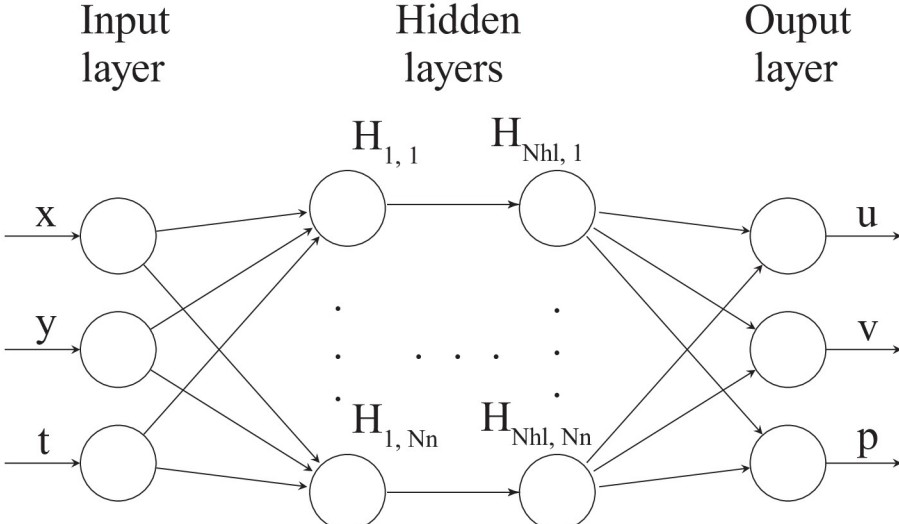

**Fig 7. Neural networks architecture used for nonlinear Biot's equations.** This figure corresponds to the top part of Fig 3. There are three inputs, $x$, $y$, and $t$, and three outputs, $u$, $v$, and $p$. The number of hidden layers, $N_{hl}$, and the number of neurons for each hidden layer, $N_n$, denote the hyperparameters.

combined optimization scheme has been reported in the literature for the case of the Navier-Stokes equations [26].

**Verification of the forward model.** Again, we apply Eq (19) and obtain:

$$MSE = MSE_b + MSE_{\Pi_u} + MSE_{\Pi_p} \tag{44}$$

where

$$MSE_b = \frac{1}{N_b}\sum_{i=1}^{N_b}(|u(x_b^i, y_b^i, t_b^i) - u^i|^2 + |v(x_b^i, y_b^i, t_b^i) - v^i|^2 \tag{45}$$
$$+ |p(x_b^i, y_b^i, t_b^i) - p^i|^2),$$

$$MSE_{\Pi_u} = \frac{1}{N_{\Pi_u}}\sum_{i=1}^{N_{\Pi_u}}|\Pi_u(x_{\Pi_u}^i, y_{\Pi_u}^i, t_{\Pi_u}^i)|^2, \tag{46}$$

and

$$MSE_{\Pi_p} = \frac{1}{N_{\Pi_p}}\sum_{i=1}^{N_{\Pi_p}}|\Pi_p(x_{\Pi_p}^i, y_{\Pi_p}^i, t_{\Pi_p}^i)|^2, \tag{47}$$

where $\{x_b^i, y_b^i, t_b^i, u^i, v^i, p^i\}_{i=1}^{N_b}$ refer to the initial and boundary training data. $\{x_{\Pi_u}^i, y_{\Pi_u}^i, t_{\Pi_u}^i\}_{i=1}^{N_{\Pi_u}}$ and $\{x_{\Pi_p}^i, y_{\Pi_p}^i, t_{\Pi_p}^i\}_{i=1}^{N_{\Pi_p}}$ specify the collocation points for $\Pi_u(x, y, t)$ and $\Pi_p(x, y, t)$, as defined in Eqs (15) and (17). Similar to the diffusivity equation case, these collocation points are sampled using the Latin Hypercube method [51]. $N_b$ denotes the number of initial and boundary training data, and $N_{\Pi_u}$ and $N_{\Pi_p}$ are the number of collocation points for $\Pi_u$ and $\Pi_p$, respectively. For the sake of simplification, in this investigation, we assume $N_{\Pi_u} = N_{\Pi_p} = N_{\Pi}$ and $\{x_{\Pi_u}^i, y_{\Pi_u}^i, t_{\Pi_u}^i\}_{i=1}^{N_{\Pi_u}} = \{x_{\Pi_p}^i, y_{\Pi_p}^i, t_{\Pi_p}^i\}_{i=1}^{N_{\Pi_p}} = \{x_{\Pi}^i, y_{\Pi}^i, t_{\Pi}^i\}_{i=1}^{N_{\Pi}}$.

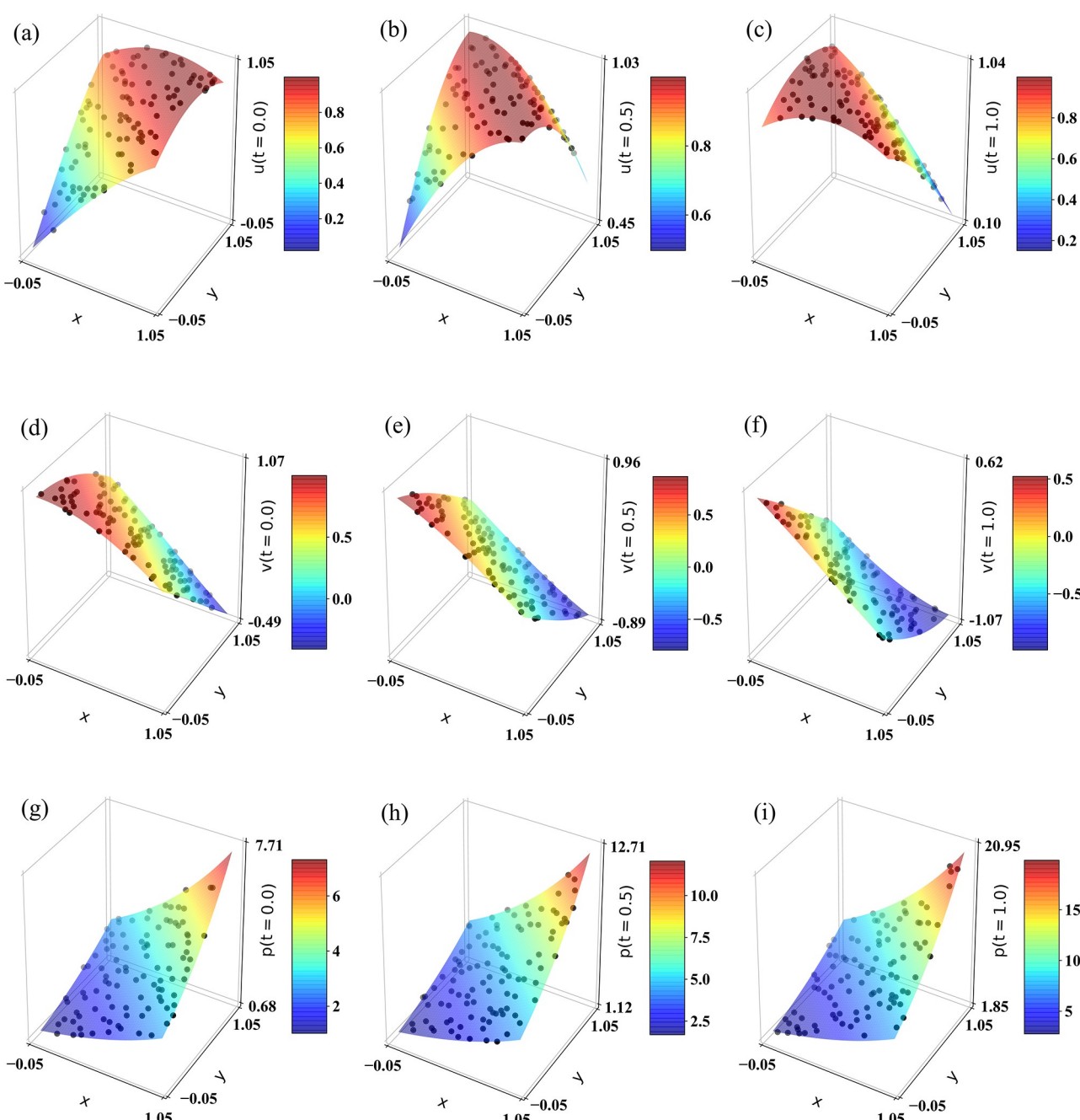

**Fig 8. Biot's equations—forward modeling: exact solutions (shown by surface plot) and 100 PINN predictions per time step using the test set (shown by black points).** The PINN was trained using $N_b = 24$, $N_\Pi = 20$, $N_{hl} = 6$, and $N_n = 5$. The top row illustrates the displacement in the x-direction, $u$, at (a) $t = 0.0$, (b) $t = 0.5$, and (c) $t = 1.0$. The middle row illustrates the displacement in the y-direction, $v$, at (d) $t = 0.0$, (e) $t = 0.5$, and (f) $t = 1.0$. The bottom row illustrates the pressure, $p$, at (g) $t = 0.0$, (h) $t = 0.5$, and (i) $t = 1.0$.

In Fig 8, we illustrate an example of exact solutions of $u$, $v$, and $p$, and compare them to the prediction values obtained from our test set using the PINN trained with $N_b = 24$, $N_\Pi = 20$, $N_{hl} = 6$, and $N_n = 5$. This figure demonstrates that the PINN provides good approximations of the exact solutions. The dependency of the prediction accuracy on $N_b$ and $N_\Pi$ with the $N_{hl}$ and $N_n$ fixed to four and ten, respectively is illustrated in Table 5. Again, to deal with the stochastic

**Table 5. Biot's equations—forward modeling: PINN performance as function of training set size.** Sum of relative $\mathcal{L}^2$ errors between the exact and predicted values of $u$, $v$, and $p$ for the validation set. The table shows the dependency on the number of the initial and boundary training data, $N_b$, and on the number of collocation points, $N_\Pi$. The hyperparemeters are fixed to 4 layers with 10 neurons per hidden layer.

**Panel A: Average over 1 realization**

| $N_\Pi$ | 2 | 5 | 10 | 20 | 40 | 80 | 160 |
|---|---|---|---|---|---|---|---|
| $N_b$ | | | | | | | |
| 2 | 0.4475 | 0.4257 | 0.5010 | 0.2865 | 0.2826 | 0.1438 | 0.1982 |
| 3 | 0.1770 | 0.1508 | 0.2464 | 0.2009 | 0.2059 | 0.1410 | 0.1751 |
| 6 | 0.4056 | 0.4615 | 0.0824 | 0.0042 | 0.1236 | 0.1327 | 0.1662 |
| 12 | 0.0412 | 0.0860 | 0.0722 | 0.0035 | 0.0005 | 0.0013 | 0.0267 |
| 24 | 0.0148 | 0.0824 | 0.0037 | 0.0019 | 0.0015 | 0.0007 | 0.0003 |
| 48 | 0.0532 | 0.0045 | 0.0027 | 0.0008 | 0.0019 | 0.0002 | 0.0099 |
| 96 | 0.0007 | 0.0005 | 0.0003 | 0.0006 | 0.0004 | 0.0001 | 0.0003 |

**Panel B: Average over 3 realizations**

| $N_\Pi$ | 2 | 5 | 10 | 20 | 40 | 80 | 160 |
|---|---|---|---|---|---|---|---|
| $N_b$ | | | | | | | |
| 2 | 0.4738 | 0.5502 | 0.5458 | 0.1493 | 0.1310 | 0.2138 | 0.1804 |
| 3 | 0.3415 | 0.1240 | 0.3578 | 0.3044 | 0.1573 | 0.1192 | 0.1562 |
| 6 | 0.5052 | 0.1423 | 0.0733 | 0.1688 | 0.2066 | 0.1158 | 0.0005 |
| 12 | 0.0551 | 0.0566 | 0.0044 | 0.0087 | 0.0472 | 0.0037 | 0.1108 |
| 24 | 0.0498 | 0.0135 | 0.0045 | 0.0005 | 0.0009 | 0.0016 | 0.0013 |
| 48 | 0.0034 | 0.0016 | 0.0246 | 0.0012 | 0.0027 | 0.0002 | 0.0015 |
| 96 | 0.0004 | 0.0002 | 0.0006 | 0.0005 | 0.0006 | 0.0001 | 0.0014 |

**Panel C: Average over 27 realizations**

| $N_\Pi$ | 2 | 5 | 10 | 20 | 40 | 80 | 160 |
|---|---|---|---|---|---|---|---|
| $N_b$ | | | | | | | |
| 2 | 0.5320 | 0.4828 | 0.4473 | 0.2458 | 0.2253 | 0.2660 | 0.3100 |
| 3 | 0.4660 | 0.4211 | 0.3503 | 0.1781 | 0.1644 | 0.1661 | 0.1870 |
| 6 | 0.4031 | 0.1765 | 0.1380 | 0.0583 | 0.0871 | 0.0852 | 0.1340 |
| 12 | 0.1086 | 0.0641 | 0.0471 | 0.0059 | 0.0169 | 0.0164 | 0.0195 |
| 24 | 0.0525 | 0.0220 | 0.0101 | 0.0142 | 0.0052 | 0.0016 | 0.0012 |
| 48 | 0.0061 | 0.0030 | 0.0013 | 0.0013 | 0.0012 | 0.0003 | 0.0009 |
| 96 | 0.0022 | 0.0005 | 0.0008 | 0.0006 | 0.0004 | 0.0007 | 0.0005 |

behavior of the neural networks, we calculate an average of the relative $\mathcal{L}^2$ error over many realizations to obtain a clear pattern; see Panels B and C of Table 5. Similar to the diffusivity equation case, we observe the accuracy of the PINN is improved when $N_b$ and $N_\Pi$ are increased. We also note that the total amount of training examples required to achieve high accuracy, i.e., an average $\mathcal{L}^2$ error less than $1.0 \times 10^{-3}$, is in the order of 100s. Again, we can illustrate the impact of the $\Pi$ terms by considering the case where we train a neural network to interpolate a feed-forward solution based on a known set of solution points with and without making use of the regularization terms. Without using $\Pi$ we obtain an average relative $\mathcal{L}^2$ error of $1.1 \times 10^{-1}$ based on 96 solution points while the average $\mathcal{L}^2$ error becomes less than $1.0 \times 10^{-3}$ when including $\Pi$.

In Table 6, we present a sensitivity analysis of $N_{hl}$ and $N_n$ with a fixed size of the training set; $N_b = 96$ and $N_\Pi = 160$. Once again, the observed trend becomes more apparent by averaging over many training realizations. We can now identify an extremum in the explored space of the hyperparameters corresponding to have $N_{hl} = 6$ and $N_n = 20$; see Panel C of Table 6. We

**Table 6. Biot's equations—forward modeling: PINN performance as function of hyperparameters.** Sum of relative $\mathcal{L}^2$ errors between the exact and predicted values of $u$, $v$, and $p$ for the validation set. The table shows the dependency on the different number of hidden layers, $N_{hl}$, and different number of neurons per layer, $N_n$. Here, the total number of training and collocation points is fixed to $N_b = 96$ and $N_\Pi = 160$, respectively.

| Panel A: Average over 1 realization | | | | | | |
|---|---|---|---|---|---|---|
| $N_n$ | 2 | 5 | 10 | 20 | 40 | 80 |
| $N_{hl}$ | | | | | | |
| 2 | 0.56506 | 0.01617 | 0.00782 | 0.00045 | 0.00021 | 0.00115 |
| 4 | 0.12080 | 0.00316 | 0.00013 | 0.00019 | 0.00020 | 0.00016 |
| 6 | 0.45949 | 0.02069 | 0.00052 | 0.00060 | 0.00010 | 0.00020 |
| 8 | 0.14333 | 0.00971 | 0.93757 | 0.00048 | 0.00034 | 0.00015 |
| 16 | 0.14052 | 0.14110 | 0.00041 | 0.00020 | 0.00020 | 0.00019 |
| 32 | 0.60485 | 0.03188 | 0.00866 | 0.00972 | 0.00028 | 0.00039 |
| Panel B: Average over 3 realizations | | | | | | |
| $N_n$ | 2 | 5 | 10 | 20 | 40 | 80 |
| $N_{hl}$ | | | | | | |
| 2 | 0.30761 | 0.01074 | 0.00346 | 0.00019 | 0.00026 | 0.00020 |
| 4 | 0.13684 | 0.01101 | 0.00032 | 0.00006 | 0.00007 | 0.00008 |
| 6 | 0.14338 | 0.04415 | 0.00018 | 0.00012 | 0.00009 | 0.00007 |
| 8 | 0.47702 | 0.01634 | 0.00020 | 0.00010 | 0.00008 | 0.00007 |
| 16 | 0.33020 | 0.13525 | 0.00081 | 0.00041 | 0.00013 | 0.00013 |
| 32 | 0.46854 | 0.61381 | 0.38437 | 0.07503 | 0.00087 | 0.00010 |
| Panel C: Average over 27 realizations | | | | | | |
| $N_n$ | 2 | 5 | 10 | 20 | 40 | 80 |
| $N_{hl}$ | | | | | | |
| 2 | 0.24203 | 0.02952 | 0.00180 | 0.00062 | 0.00021 | 0.00023 |
| 4 | 0.39596 | 0.06005 | 0.00368 | 0.00039 | 0.00010 | 0.00014 |
| 6 | 0.32610 | 0.07829 | 0.00018 | 0.00009 | 0.00010 | 0.00014 |
| 8 | 0.42070 | 0.12314 | 0.00037 | 0.00013 | 0.00010 | 0.00012 |
| 16 | 0.46364 | 0.12222 | 0.09613 | 0.05190 | 0.00012 | 0.00011 |
| 32 | 0.41372 | 0.38439 | 0.38473 | 0.03998 | 0.05080 | 0.01052 |

then apply all of the 27 PINN networks trained with that choice of hyperparameters to the test set. We obtain the average $\mathcal{L}^2$ error of the test set to be $9.05 \times 10^{-5}$, which is comparable to that of the validation set, $9.08 \times 10^{-5}$.

In Fig 9, we show the behavior of the different loss terms as function of the training iterations, when using $N_{hl} = 6$, $N_n = 20$, $N_b = 96$, and $N_\Pi = 160$. Similar to the diffusivity equation, we observe that $MSE_b$ is generally higher than $MSE_{\Pi_u}$ and $MSE_{\Pi_p}$. Furthermore, $MSE_{\Pi_u}$ and $MSE_{\Pi_p}$ are comparable. Unlike the diffusivity equation case (see Fig 5), we observe minor oscillations of $MSE$, $MSE_b$, $MSE_{\Pi_u}$, and $MSE_{\Pi_p}$ during convergence.

**Inverse model of Biot's equations.** We rewrite Eq (15) to the parametrized form as:

$$\Pi_u = \nabla \cdot [2\theta_1 \varepsilon(u) + \theta_2 uI] - \theta_3 \nabla \cdot pI - f \text{ in } \Omega \times \mathbb{T}, \tag{48}$$

and Eq (17) as:

$$\Pi_p = \theta_4 \frac{\partial p}{\partial t} + \theta_3 \frac{\partial \nabla \cdot u}{\partial t} - \theta_5 \nabla \cdot e^{\varepsilon_v}(\nabla p - \rho\mathbf{g}) - g \text{ in } \Omega \times \mathbb{T}, \tag{49}$$

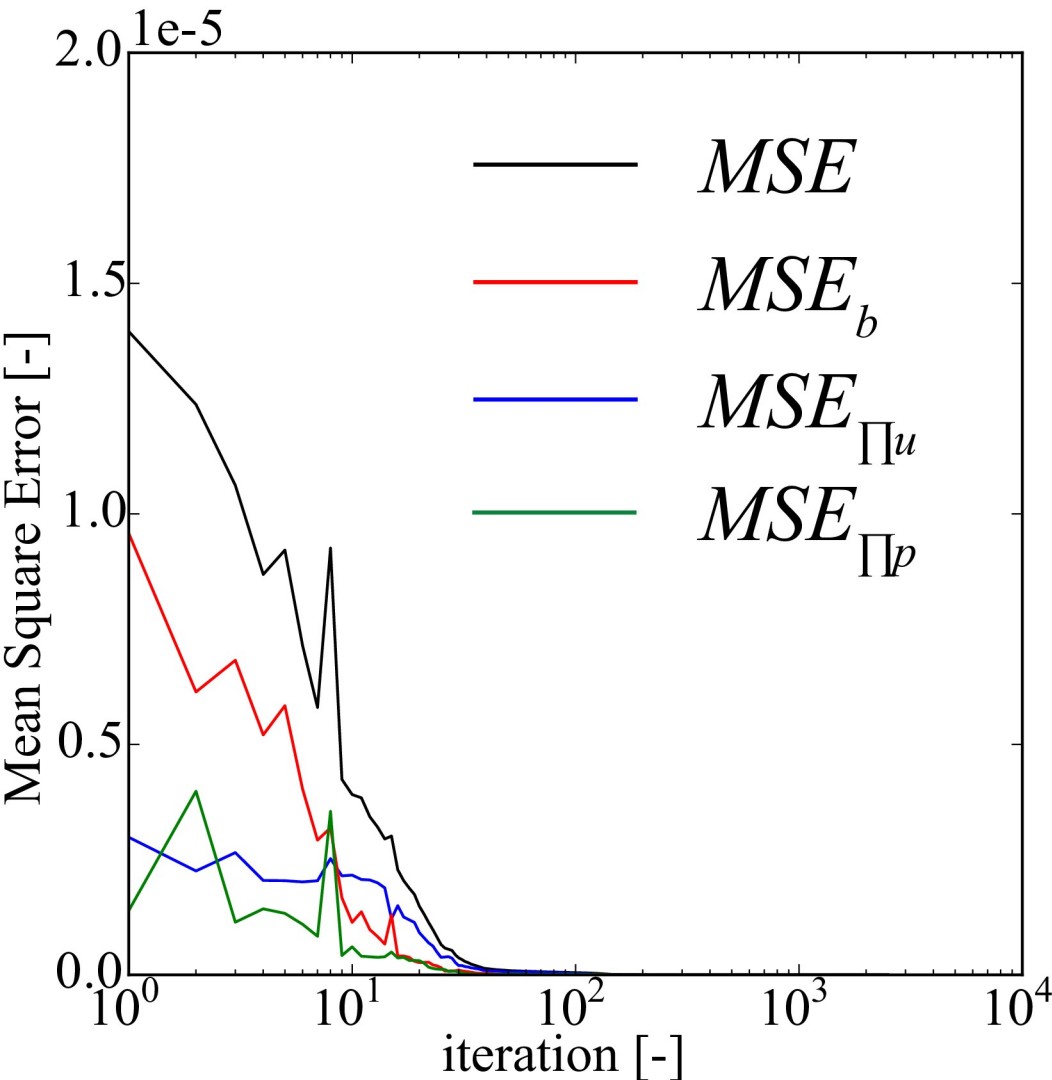

**Fig 9. Biot's equations—forward modeling: mean square training error plot.** This model uses $N_{hl}$ = 6, $N_n$ = 20, $N_b$ = 96, and $N_\Pi$ = 160. $MSE$, $MSE_b$, $MSE_{\Pi_u}$, and $MSE_{\Pi_p}$ are calculated using Eqs 44, 45, 46, and 47, respectively.

where

$$\theta_1 = \mu_l, \quad \theta_2 = \lambda_l, \quad \theta_3 = \alpha, \quad \theta_4 = \phi c_f + \frac{\alpha - \phi}{K_s}, \text{ and } \quad \theta_5 = \kappa_0. \tag{50}$$

The $\Pi$ terms now have five additional unknown parameters, $\theta_1$, $\theta_2$, $\theta_3$, $\theta_4$, and $\theta_5$ that along with the weights and biases of the neural network are adjusted during the training of the network. Once again we apply Eq (19) and obtain:

$$MSE = MSE_{tr} + MSE_{\Pi_u} + MSE_{\Pi_p} \tag{51}$$

where

$$MSE_{tr} = \frac{1}{N_{tr}} \sum_{i=1}^{N_{tr}} (|u(x_{tr}^i, y_{tr}^i, t_{tr}^i) - u^i|^2 + |v(x_{tr}^i, y_{tr}^i, t_{tr}^i) - v^i|^2$$
$$+ |p(x_{tr}^i, y_{tr}^i, t_{tr}^i) - p^i|^2), \tag{52}$$

$$MSE_{\Pi_u} = \frac{1}{N_{\Pi_u}} \sum_{i=1}^{N_{\Pi_u}} |\Pi_u\left(x_{\Pi_u}^i, y_{\Pi_u}^i, t_{\Pi_u}^i\right)|^2, \tag{53}$$

and

$$MSE_{\Pi_p} = \frac{1}{N_{\Pi_p}} \sum_{i=1}^{N_{\Pi_p}} |\Pi_p\left(x_{\Pi_p}^i, y_{\Pi_p}^i, t_{\Pi_p}^i\right)|^2, \tag{54}$$

where $\{x_{tr}^i, y_{tr}^i, t_{tr}^i, u^i, v^i, p^i\}_{i=1}^{N_{tr}}$ refer to the set of training data. In contrast to the forward model, we can apply the training points as collocation points when calculating the terms given by Eqs (48) and (49). To recap, the $N_{tr}$ denotes the number of training data. Similar to the forward

**Table 7. Biot's equations—inverse modeling: PINN performance as function of hyperparameters.** Relative $\mathcal{L}^2$ error of $p$, $u$, and $v$ and percentage error of $\theta_1$, $\theta_2$, $\theta_3$, $\theta_4$, and $\theta_5$ for different number of hidden layers, $N_{hl}$, and different number of neurons per layer, $N_n$. The $N_{tr}$ is fixed at 250. Note that we pick the optimal hyperparameters, i.e., $N_{hl} = 6$ and $N_n = 20$ from the sensitivity analysis of the forward model. Results shown in this table are an average over 27 realizations.

| | $N_n$ | 2 | 5 | 10 |
|---|---|---|---|---|
| | $N_{hl}$ | | | |
| $p$ | 4 | 0.00006 | 0.00004 | 0.00008 |
| | 6 | 0.00009 | 0.00004 | 0.00006 |
| | 8 | 0.00008 | 0.00006 | 0.00005 |
| $u$ | 4 | 0.00019 | 0.00016 | 0.00047 |
| | 6 | 0.00051 | 0.00013 | 0.00036 |
| | 8 | 0.00027 | 0.00036 | 0.00034 |
| $v$ | 4 | 0.00042 | 0.00045 | 0.00098 |
| | 6 | 0.00114 | 0.00038 | 0.00074 |
| | 8 | 0.00060 | 0.00079 | 0.00071 |
| $\theta_1$ | 4 | 0.20180 | 0.28665 | 1.06223 |
| | 6 | 0.72512 | 0.16844 | 0.97916 |
| | 8 | 0.30534 | 0.70613 | 0.99229 |
| $\theta_2$ | 4 | 0.65507 | 0.66560 | 3.56789 |
| | 6 | 2.23747 | 0.23053 | 2.80360 |
| | 8 | 1.21152 | 1.47513 | 2.45137 |
| $\theta_3$ | 4 | 0.02968 | 0.05016 | 0.23851 |
| | 6 | 0.33292 | 0.02935 | 0.10100 |
| | 8 | 0.22593 | 0.11796 | 0.07371 |
| $\theta_4$ | 4 | 0.04834 | 0.06162 | 0.19135 |
| | 6 | 0.18104 | 0.04666 | 0.13785 |
| | 8 | 0.10711 | 0.16124 | 0.12127 |
| $\theta_5$ | 4 | 0.19710 | 0.21622 | 1.10700 |
| | 6 | 1.24043 | 0.17633 | 0.60595 |
| | 8 | 0.81199 | 0.50331 | 0.39558 |

model, all physical constants are set to one, i.e., $\theta_1$, $\theta_2$, $\theta_3$, $\theta_4$, and $\theta_5$ are equal to one. Note that these $\theta$ are constant throughout the domain.

We use the optimal hyperparameters, i.e., $N_{hl}$ = 6 and $N_n$ = 20 from the sensitivity analysis of the forward model (see Panel C of Table 6). In Table 7, we can observe that this choice also yields the least $\mathcal{L}^2$ error with respect to both $p$, $u$, and $v$ and the lowest percentage error for the unknown physical parameters ($\theta_1$, $\theta_2$, $\theta_3$, $\theta_4$, and $\theta_5$). Moreover, we observe that the $\mathcal{L}^2$ error of the output space and percentage error of the unknown physical parameters are not much different with different combinations of the hyperparameters.

The performance of the PINN model as a function of the number of training examples $N_{tr}$ is depicted in Fig 10. We can observe that the stochastic variations in the estimated values of $\theta_1$, $\theta_2$, $\theta_3$, $\theta_4$ and $\theta_5$, in general, are reduced the more training examples we apply. Moreover, the relative $\mathcal{L}^2$ errors of $u$, $v$, and $p$ are always less than 0.01%. Using in the order of 1000 examples, the average estimation error of the physical parameters is in the order of 1%, but as with the diffusivity case, there is a large variation between the trained PINN models. Within the 27 realizations of the trained PINN models with $N_{tr}$ = 1000 the percentage errors of $\theta_1$, $\theta_2$, $\theta_3$, $\theta_4$ and $\theta_5$ varied between 0.05, 0.02, 0.03, 0.04, and 0.03 and 2.51, 6.14, 4.75, 8.12, and 3.60, respectively. Having 1000 training examples in actual cases, i.e., lab experiments or field observations, is realistic. Hence, also for the Biot's equations, we observe the feasibility of the PINN model to handle the inverse problem by estimating the unknown physical parameters using a reasonably sized data set.

Next, we perform a systematic study of the effect of noise in data, which is created utilizing Eq (35). The results are presented in Table 8 for $\theta_1$, $\theta_2$, $\theta_3$, $\theta_4$, and $\theta_5$. Note that these results are an average over ten realizations. We can observe that the percentage error of $\theta_1$, $\theta_2$, $\theta_3$, $\theta_4$, and $\theta_5$ increase along with the noise level ($\epsilon$), but as $N_{tr}$ is increased, the impact of the noise is reduced.

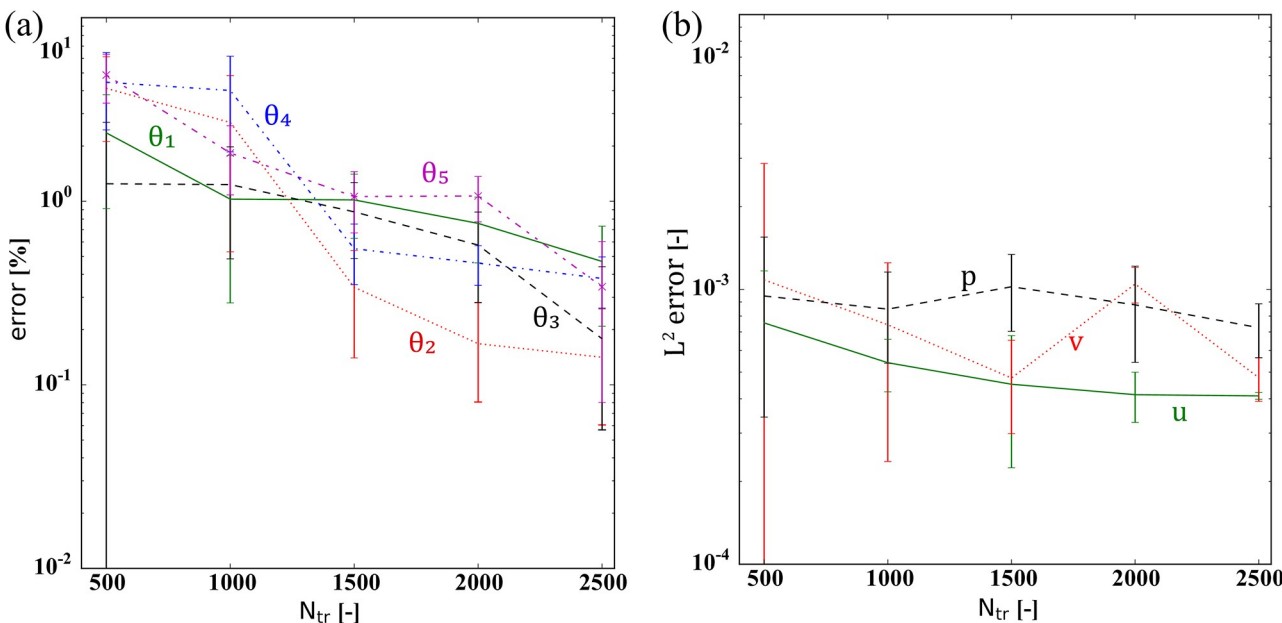

**Fig 10. Biot's equations—inverse modeling: PINN performance as function of training set size.** This figure shows the estimated error dependency on the amount of training data, $N_{tr}$. (a) $\theta_1$, $\theta_2$, $\theta_3$, $\theta_4$, and $\theta_5$ and (b) $u$, $v$, and $p$. The error bars show mean and standard derivation ($\pm$ 1 SD) based on 27 realizations. The reported error values of $\theta_1$, $\theta_2$, $\theta_3$, $\theta_4$, and $\theta_5$ are percentage errors while the relative $\mathcal{L}^2$ error is shown for $u$, $v$, and $p$. Note that there is no noise in this investigation.

**Table 8. Biot's equations—inverse modeling: PINN performance as function of noise.** This figure shows the average percentage errors of $\theta_1$, $\theta_2$, $\theta_3$, $\theta_4$, and $\theta_5$ for different numbers of training data $N_{tr}$ corrupted by different noise levels ($\epsilon$). Here, the neural network architecture is kept fixed to 6 layers and 20 neurons per layer. These results are an average over 10 realizations.

| Noise ($\epsilon$) | | 0% | 1% | 5% | 10% |
|---|---|---|---|---|---|
| $N_{tr}$ | | | | | |
| $\theta_1$ | 1000 | 0.91 | 6.40 | 7.15 | 29.55 |
| | 1500 | 0.98 | 4.64 | 5.70 | 14.86 |
| | 2000 | 0.67 | 2.17 | 5.26 | 15.95 |
| | 2500 | 0.52 | 3.87 | 5.33 | 12.24 |
| | 5000 | 0.30 | 1.39 | 3.32 | 7.60 |
| $\theta_2$ | 1000 | 1.48 | 10.26 | 12.30 | 17.50 |
| | 1500 | 0.84 | 6.95 | 5.44 | 14.53 |
| | 2000 | 1.37 | 4.38 | 6.73 | 13.48 |
| | 2500 | 0.80 | 3.95 | 2.91 | 6.31 |
| | 5000 | 0.51 | 1.97 | 2.90 | 3.14 |
| $\theta_3$ | 1000 | 1.19 | 2.85 | 5.06 | 8.10 |
| | 1500 | 0.87 | 1.32 | 2.52 | 4.80 |
| | 2000 | 0.58 | 1.35 | 2.49 | 4.11 |
| | 2500 | 0.11 | 0.38 | 1.54 | 4.12 |
| | 5000 | 0.10 | 0.24 | 1.72 | 2.36 |
| $\theta_4$ | 1000 | 0.46 | 4.70 | 8.35 | 13.75 |
| | 1500 | 0.43 | 3.01 | 5.22 | 8.53 |
| | 2000 | 0.23 | 2.99 | 4.12 | 10.08 |
| | 2500 | 0.24 | 0.64 | 5.88 | 9.34 |
| | 5000 | 0.20 | 0.45 | 1.62 | 6.53 |
| $\theta_5$ | 1000 | 3.46 | 9.22 | 16.15 | 24.52 |
| | 1500 | 2.70 | 5.64 | 14.16 | 19.75 |
| | 2000 | 0.44 | 3.76 | 11.56 | 17.55 |
| | 2500 | 0.66 | 2.23 | 6.82 | 7.09 |
| | 5000 | 0.24 | 1.28 | 4.26 | 5.91 |

As the noise ($\epsilon$) is increased, the error increases as expected.

All calculations were carried out using a XeonE5_2660v3 processor with a single thread. As an example of the Biot's equations, the CPU time for training the neural networks using $N_{tr} = 10000$ and 15000 with no noise are 128037 seconds and 186154 seconds, respectively. Note that the reported values are obtained from the model trained using the combined ADAM and L-BFGS. Using L-BFGS alone, the CPU times of model with $N_{tr} = 10000$ and 15000 are 222681 and 294768 seconds, respectively.

## Conclusion

This paper studies the application of physics-informed neural networks (PINN) for solving the nonlinear diffusivity and Biot's equations in the context of forward and inverse modelings. The following conclusions are drawn:

- PINN can be used to solve the forward modeling problem for the nonlinear diffusivity and Biot's equations, at least for the type of geometries considered in this paper. The displacement and pressure variables of our test sets could be predicted with an average $\mathcal{L}^2$ error of $9.05 \times 10^{-5} \pm 3.1 \times 10^{-4}$ based on 27 realizations.

- For the inverse modeling cases, PINN can predict all of the unknown physical parameters with an average percentage error of around 1%; however, the stochastic variations from one PINN implementation to the next is quite large. Using, for instance, 1000 training examples in the Biot's equation case, the percentage error of the estimated physical parameters over 27 PINN models could vary from 0.02 to 8.12. Increasing the number of training examples reduces this problem. Still, our results indicate it would be essential to do an average over PINN models with different random initialization of the weights and biases. This process may lead to the requirement of more processing power. This challenge might be even higher when applying PINN to more complex geometries and heterogeneous materials.

- For the inverse modeling, PINN is tolerant to a noise level up to 5% (the estimation error of physical parameters is approximately less than 15%.). Again, this requires that one does an average over several PINN realizations. As expected, the result improves when the number of training examples is increased.

- We have presented arguments on why the hyperparameters selection process for the forward case is likely to be applicable to the inverse case. For the cases considered here, this was confirmed experimentally. However, this should be explored in more detail by investigating the use of PINN for other types of nonlinear partial differential equations.

Finally, in terms of future work, the capability of the physics-informed neural networks should be tested in the case where the input data is incomplete, i.e., $u$ and $p$ are not available at the same spatial and temporal coordinates. Besides, one could investigate the potential benefits of training networks using mini-batches. Moreover, smarter initialization of the weights and biases (based on transfer learning principles) could potentially be employed to increase the speed and accuracy of the training procedure [56].

## Author Contributions

**Conceptualization:** Teeratorn Kadeethum, Thomas M. Jørgensen, Hamidreza M. Nick.

**Formal analysis:** Teeratorn Kadeethum.

**Funding acquisition:** Hamidreza M. Nick.

**Software:** Teeratorn Kadeethum.

**Supervision:** Thomas M. Jørgensen, Hamidreza M. Nick.

**Validation:** Thomas M. Jørgensen, Hamidreza M. Nick.

**Writing – original draft:** Teeratorn Kadeethum, Thomas M. Jørgensen.

**Writing – review & editing:** Teeratorn Kadeethum, Thomas M. Jørgensen, Hamidreza M. Nick.

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
