## [Decision Letter · Decision Letter 0]

22 Oct 2019

PONE-D-19-24031

Physics-informed Neural Networks for Solving Nonlinear Diffusivity and Biot's equations

PLOS ONE

Dear Mr. Kadeethum,

Thank you for submitting your manuscript to PLOS ONE. After careful consideration, we feel that it has merit but does not fully meet PLOS ONE’s publication criteria as it currently stands. Therefore, we invite you to submit a revised version of the manuscript that addresses the points raised during the review process.

We would appreciate receiving your revised manuscript by Dec 06 2019 11:59PM. To enhance the reproducibility of your results, we recommend that if applicable you deposit your laboratory protocols in protocols.io, where a protocol can be assigned its own identifier (DOI) such that it can be cited independently in the future. For instructions see: http://journals.plos.org/plosone/s/submission-guidelines#loc-laboratory-protocols

We look forward to receiving your revised manuscript.

Kind regards,

Fang-Bao Tian

Academic Editor

PLOS ONE

Journal Requirements:

3. In your data statement you state:

'No, some restrictions will apply'. The source codes used to support the findings of this study are developed on the open-source platform and may be released upon application to the Danish Hydrocarbon Research and Technology Centre, which can be contacted at dhrtc@dtu.dk.'

Please clarify the nature of these restrictions.

Additional Editor Comments (if provided):

Reviewers' comments:

Reviewer's Responses to Questions

**Comments to the Author**

1. Is the manuscript technically sound, and do the data support the conclusions?

Reviewer #1: Partly

2. Has the statistical analysis been performed appropriately and rigorously? 

Reviewer #1: N/A

3. Have the authors made all data underlying the findings in their manuscript fully available?

Reviewer #1: No

4. Is the manuscript presented in an intelligible fashion and written in standard English?

Reviewer #1: Yes

5. Review Comments to the Author

Reviewer #1: Please see the attached pdf document for all my comments, issues and a recommendation to the authors. I could not find any additional data provided by the authors, or supporting information.

6. PLOS authors have the option to publish the peer review history of their article (what does this mean?). If published, this will include your full peer review and any attached files.

Reviewer #1: No

---

## [Author Response · Author response to Decision Letter 0]

25 Feb 2020

Please find responses in the attached document.

---

## [Editor Report · Decision Letter 1]

21 Apr 2020

Physics-informed Neural Networks for Solving Nonlinear Diffusivity and Biot's equations

PONE-D-19-24031R1

Dear Dr. Kadeethum,

We are pleased to inform you that your manuscript has been judged scientifically suitable for publication and will be formally accepted for publication once it complies with all outstanding technical requirements.

With kind regards,

Fang-Bao Tian

Academic Editor

PLOS ONE
---

## [Editor Report · Acceptance letter]

23 Apr 2020

PONE-D-19-24031R1 

Physics-informed Neural Networks for Solving Nonlinear Diffusivity and Biot's equations 

Dear Dr. Kadeethum:

I am pleased to inform you that your manuscript has been deemed suitable for publication in PLOS ONE. Congratulations! Your manuscript is now with our production department. 

With kind regards,

on behalf of

Dr. Fang-Bao Tian 

Academic Editor

PLOS ONE